# GSFIXER: IMPROVING 3D GAUSSIAN SPLATTING WITH REFERENCE-GUIDED VIDEO DIFFUSION PRIORS

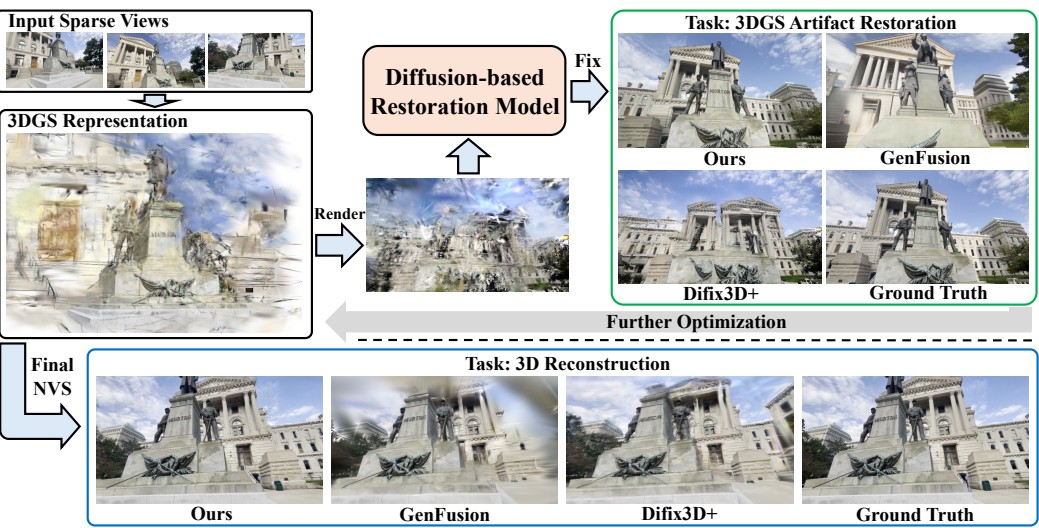

Figure 1: We introduce GSFixer, a framework capable of improving 3DGS in both artifact restoration (top) and 3D reconstruction (bottom) under sparse-view settings. Recent generative methods struggle with maintaining consistency between generated and input views. GSFixer guides the video diffusion model conditioned on both 3D geometric and 2D semantic signals to enhance consistency in novel view restoration, thereby improving 3D reconstruction quality.

## ABSTRACT

Reconstructing 3D scenes using 3D Gaussian Splatting (3DGS) from sparse views is an ill-posed problem due to insufficient information, often resulting in noticeable artifacts. While recent approaches have sought to leverage generative priors to complete information for under-constrained regions, they struggle to generate content that remains consistent with input observations. To address this challenge, we propose **GSFixer**, a novel framework designed to improve the quality of 3DGS representations reconstructed from sparse inputs. The core of our approach is the reference-guided video restoration model, built upon a DiT-based video diffusion model trained on paired artifact 3DGS renders and clean frames with additional reference-based conditions. Considering the input sparse views as references, our model integrates both 2D semantic and 3D geometric features of reference views extracted from the visual geometry foundation model, enhancing the semantic coherence and 3D consistency when fixing artifact novel views. Furthermore, we introduce a reference-guided trajectory sampling strategy that ensures both angular coverage and view quality, further enhancing reconstruction fidelity. Considering the lack of suitable benchmarks for 3DGS artifact restoration evaluation, we present DL3DV-Res which contains artifact frames rendered using low-quality 3DGS. Extensive experiments demonstrate our GSFixer outperforms current state-of-the-art methods in 3DGS artifact restoration and sparse-view 3D reconstruction. The Project will be made public.

# 1 INTRODUCTION

3D reconstruction and novel view synthesis (NVS) are fundamental tasks in computer vision and graphics, with wide-ranging real-world applications in virtual reality, autonomous driving, and robotics. Recently, 3D Gaussian Splatting (3DGS) (Kerbl et al., 2023) has achieved impressive results in both reconstruction quality and rendering efficiency when dense input views are available. However, its performance degrades significantly in sparse-view settings, where limited viewpoint information leads to under-constrained 3D representations. In such cases, 3DGS often suffers from severe artifacts, including distorted geometric structures and incomplete reconstructions, particularly in less-observed regions or extreme novel viewpoints. These limitations hinder its applicability in real-world scenarios where acquiring dense multi-view data is challenging.

To alleviate the limitations, some previous regularization methods have been proposed to introduce additional constraints into the 3DGS optimization process, such as monocular depth (Li et al., 2024; Zhu et al., 2024), frequency smoothness (Zhang et al., 2024), and random dropout (Xu et al., 2025). While these approaches can help prevent 3DGS representations from overfitting to sparse input views, they often remain sensitive to noise and yield only marginal improvements in NVS rendering quality. Inspired by the success of ReconFusion (Wu et al., 2024), which introduces diffusion model into NeRF (Mildenhall et al., 2020) optimization, more recent studies (Liu et al., 2024b;a; Wu et al., 2025a;b) explore incorporating 3DGS optimization with powerful generative priors from diffusion models. These strong priors enable the correction of spurious geometry or the inpainting of plausible content in novel views. However, these diffusion-based approaches struggle to preserve faithful alignment with input views, especially under large viewpoint extrapolation.

We argue that this limitation arises because current methods lack a deep understanding of the given sparse views, as they treat the artifact frames in the same manner as the clean reference Wu et al. (2025b) in the video diffusion model or utilize the image diffusion model, causing temporal inconsistency (Wu et al., 2025a). Both previous considerations have limited specific designs to utilize the input features and fail to capture both semantic and geometric correctness from the generative prior of the video diffusion model. Yet, semantic consistency and geometry coherence are two essential properties for high-fidelity NVS, as inconsistencies in restored frames further influence the reconstruction accuracy of the 3DGS, as shown in Fig. 1.

Inspired by the above motivation, we propose **GSFixer**, a novel generative reconstruction framework built upon a DiT-based video diffusion model (Yang et al., 2024) (VDM) for progressive 3DGS reconstruction. The proposed VDM is trained on paired 3DGS artifact renders and clean frames, with additional multi-aspect extraction features as reference. To introduce multiple knowledge from the original clean input image, we extract the features of the reference view via semantic visual encoders (*e.g.*, DINOv2 (Oquab et al., 2024)) as a semantic signal to ensure semantic consistency between the fixed novel views and the input observations. To further enforce multi-view consistency, we incorporate feed-forward 3D reconstruction networks (*e.g.*, VGGT (Wang et al., 2025)) for better geometric understanding. Moreover, we introduce a reference trajectory sampling strategy into the iterative generative optimization process to fix artifacts in novel views, effectively balancing angular coverage and restoration quality. In addition, to facilitate the evaluation of artifact removal capabilities of generative methods, we present DL3DV-Res, a benchmark comprising artifact-ridden frames rendered from low-quality 3DGS representations. Empirical results demonstrate that our GSFixer achieves superior 3DGS artifact restoration and sparse-view 3D reconstruction performance on various challenging scenes.

Our main contributions are summarized as follows:

- We introduce GSFixer, a novel generative reconstruction framework tailored for improving the quality of 3DGS representations. It integrates a reference-guided video restoration model along with a reference-guided trajectory sampling strategy.

- We propose to incorporate video diffusion model conditioned on both 2D semantic and 3D geometric features extracted from the reference views. This effectively guides the restoration of artifact novel views, achieving both semantically coherent and 3D consistent results.

- We present DL3DV-Res benchmark for evaluating the performance of generative models in 3DGS artifact restoration. Extensive experiments demonstrate that our GSFixer outperforms existing baselines in 3DGS artifact restoration and sparse-view 3D reconstruction.

## 2 RELATED WORKS

**Regularization Sparse-view Novel View Synthesis.**  Neural Radiance Fields (NeRF) and 3DGS have revolutionized the field of NVS by leveraging advances in neural rendering. While NeRF and 3DGS achieve remarkable photorealistic novel-view synthesis results given dense input views, they often suffer from severe overfitting issues and significant artifacts when training with sparse views. Previous works have attempted to address this limitation by introducing additional constraints and regularization into the per-scene 3D optimize process, such as depth supervision (Deng et al., 2022; Roessle et al., 2022; Wang et al., 2023; Li et al., 2024; Zhu et al., 2024), normal consistency (Yu et al., 2022; Seo et al., 2023), smoothness priors (Niemeyer et al., 2022; Yang et al., 2023; Zhang et al., 2024), semantic coherence (Jain et al., 2021; Truong et al., 2023), and random dropout strategy (Xu et al., 2025). However, these approaches often yield marginal improvements and remain sensitive to some specific scene data.

**Conditional Video Diffusion Models.**  Recently, generative models (Rombach et al., 2022; Liu et al., 2023; Blattmann et al., 2023; Chen et al., 2023; 2024; Xing et al., 2024; Yang et al., 2024) have advanced rapidly and demonstrate impressive capabilities in high-quality visual content generation. Building on this progress, recent efforts have extended video diffusion models with various control signals, such as depth (Hu et al., 2025; Zhao et al., 2024), trajectory (Niu et al., 2024; YU et al., 2025), audio (Cui et al., 2025; Kong et al., 2025), camera parameters (Wang et al., 2024b; Zhang et al., 2025), and point cloud renderings (Yu et al., 2024; Gu et al., 2025). The additional control signals enable the diffusion models to generate coherent content with the user's inputs. Our work builds upon a video diffusion framework that leverages 3DGS renderings and features from clean reference frames as condition signals to guide the restoration of artifacts in novel views.

**Generative Sparse-view Novel View Synthesis.**  Generative models have demonstrated a strong ability to inpaint plausible content in unobserved regions and restore degraded areas with high visual fidelity. Recent approaches (Wu et al., 2024; Liu et al., 2024b;a; Wu et al., 2025a; Bao et al., 2025; Wu et al., 2025b) have made notable progress in the sparse-view NVS task by leveraging priors from generative models. For instance, ReconFusion (Wu et al., 2024) combines image diffusion with PixelNeRF (Yu et al., 2021) as a condition to optimize NeRF representations. Similarly, more recent works (Liu et al., 2024b;a; Wu et al., 2025a; Bao et al., 2025; Wu et al., 2025b; Fischer et al., 2025) utilize diffusion priors to enhance low-qualtiy 3DGS representations. ReconX (Liu et al., 2024a) leverages explicit pointmap output of DUSt3R (Wang et al., 2024a) as 3D condition, which may suffer from noise or structural errors. Concurrently with our work, FlowR (Fischer et al., 2025) enhances novel rendered from the 3D representation via flow-matching. Our work is closely related to 3DGS-Enhancer (Liu et al., 2024b), GenFusion (Wu et al., 2025b) and DIFIX3D+ (Wu et al., 2025a), all of which finetune diffusion models to correct artifacts in novel views. While following a similar direction, our method differs in two key aspects: (i) we finetune a DiT-based video diffusion model controlled with additional reference view conditions, and (ii) we incorporate both 2D semantic features and 3D geometric features extracted from visual geometry foundation models to guide the video restoration process, effectively fixing the artifacts in novel views.

## 3 METHODS

Given $K$ sparse-view RGB images $\{I_i\}_{i=1,...,K}$ and corresponding camera poses $\{P_i\}_{i=1,...,K}$, our goal is to build 3DGS from these inputs and improve its reconstruction quality via the diffusion-based novel view artifacts restoration, as shown in Fig. 2. To achieve this, we propose a reference-guided video restoration model that refines artifact-ridden novel views and distills the restored results back into the 3DGS representation, enabling progressive reconstruction enhancement. The restoration model leverages additional semantic and geometric features extracted from the reference views. These features are subsequently projected, fused, and injected into a video diffusion process to guide the restoration, preserving faithful semantic alignment and geometric consistency with the input views (Sec. 3.1). Moreover, we introduce a reference-guided trajectory sampling strategy (RGT) within the iterative generative optimization to further enhance the quality. (Sec. 3.2). *Please refer to the appendix for preliminaries on 3D Gaussian Splatting and video diffusion models.*

### 3.1 REFERENCE-GUIDED VIDEO RESTORATION MODEL

Recent works (Liu et al., 2024b; Wu et al., 2025b) have introduced video diffusion priors to restore artifact-prone novel views into clean frames. However, the generated frames often lack visual and

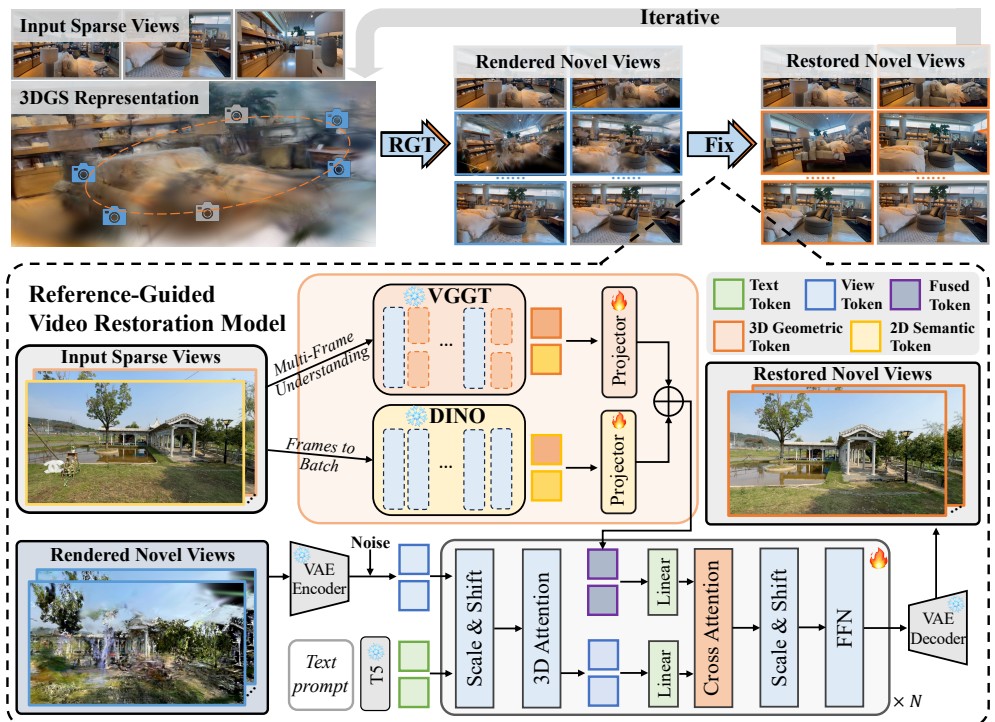

Figure 2: **Pipeline of GSFixer.** Given sparse-view images and their corresponding low-quality 3DGS representation, we render artifact-prone novel views between two reference views along a reference-guided trajectory. These novel views are fed into the reference-guided video restoration model to correct artifacts, and the fixed novel views are then distilled back into the 3DGS representation to improve its quality. The restoration network is finetuned from CogVideoX and trained on paired artifact-ridden 3DGS renders and ground truth frames. It is additionally conditioned on 3D geometric tokens and 2D semantic tokens extracted from the reference views using pretrained VGGT and DINOv2 encoder, respectively.

3D consistency with the input sparse views, leading to suboptimal 3D reconstruction performance. To address this, considering the artifacts finally lie in the 2D image space and are caused by suboptimal 3DGS representations in 3D space, we propose injecting both 2D semantic and 3D geometric signals of reference views to control the video diffusion process, ensuring semantic and multi-view consistency in restoring the artifact-prone novel views.

As illustrated in Fig. 2, our reference-guided video restoration model is built upon DiT-based video diffusion model CogVideoX (Yang et al., 2024), an image-to-video diffusion model capable of animating an input image to generate a video. We utilize its 3D Variational Autoencoder (VAE) for video compression and decompression. Additionally, we use BLIP (Li et al., 2022) and T5 (Raffel et al., 2020) encoder to caption video and extract the text tokens $t_i^{1D}$.

In our video-to-video restoration task, which aims to restore artifact frames into clean ones, we replace the original image condition with the 3DGS renders $\{I_n^{nov}\}_{n=1,...,N}$ between two reference views $\{I_i, I_j\}$. Both the artifact frames and reference views are encoded using the 3D VAE encoder $\mathcal{E}$, concatenated with per-frame initial noise, and projected into view tokens $t^{view}$. These view tokens are then combined with text tokens $t^{text}$ and fed into the DiT blocks for denoising.

**Reference-based Conditions.** Given a set of sparse input images, we select two images $\{I_i, I_j\}$ as reference views and generate a set of novel views $\{I_n^{nov}\}_{n=1,...,N}$ along arbitrary trajectories between them. We aim to learn a conditional $\boldsymbol{x} \sim p(\boldsymbol{x}|I^{nov}, \{I_i, I_j\})$ to guide the video diffusion model process toward consistent and artifact-free outputs. To incorporate 2D semantic guidance, we employ the pretrained DINOv2 (Oquab et al., 2024) model as a 2D visual tokenizer. For each reference view $I_i$, DINOv2 encoder $\mathcal{E}_{2D}$ divides it into $L$ patches and extracts robust 2D visual features as a sequence of tokens:

$$t_i^{2D} = \mathcal{E}_{2D}(I_i), t_i^{2D} \in \mathbb{R}^{L \times C}, \qquad (1)$$

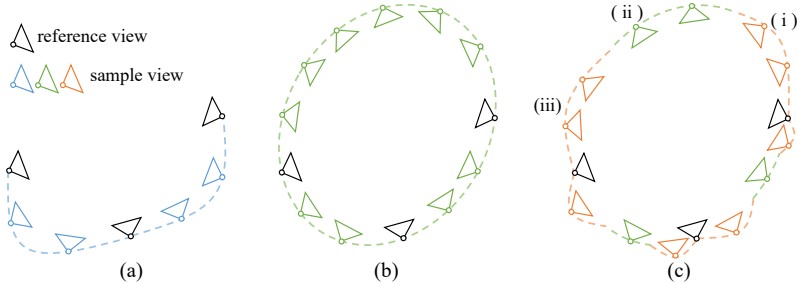

Figure 3: **Illustration of different trajectories.** (a) Interpolation trajectory: blue curve. (b) Ellipse trajectory: green curve. (c) Reference-guided trajectory: orange and green curve.

where $t_i^{2D}$ denotes the resulting 2D semantic tokens, with $L$ and $C$ representing the number of tokens and the feature dimension, respectively.

To further control our reference-guided video diffusion model with 3D geometric priors, we adopt the pretrained Visual Geometry Grounded Transformer (VGGT) (Wang et al., 2025) as a 3D geometric tokenizer. VGGT employs a 3D geometric encoder $\mathcal{E}_{3D}$, which composes several frame-wise and global self-attention layers, to encode multi-view reference views $\{I_i, I_j\}$:

$$t_i^{3D}, t_j^{3D} = \mathcal{E}_{3D}(I_i, I_j), t_i^{3D} \in \mathbb{R}^{L \times C}, t_j^{3D} \in \mathbb{R}^{L \times C}, \tag{2}$$

where $t_i^{3D}, t_j^{3D}$ represent the extracted 3D geometric tokens, with $L$ and $C$ representing the number of tokens and the feature dimension, respectively. VGGT leverages these tokens to produce all key 3D attributes of a scene, including camera parameters, depth maps, point maps, and 3D point tracks through various prediction heads. Consequently, the 3D geometric tokens include rich and robust 3D geometric priors of the scene.

**Reference-guided Generation.** As previously mentioned, the fixed frames may still lack visual and 3D consistency with the reference images. To address this, we fuse 3D geometric tokens and 2D semantic tokens of reference views to fusion tokens as an additional condition:

$$t_i^{fusion} = Projector_{3D}(t_i^{3D}) + Projector_{2D}(t_i^{2D}), \tag{3}$$

where $Projector_{3D}$ and $Projector_{2D}$ are implemented using linear and normalization layers.

To inject the reference fusion tokens to control the video diffusion process, we augment each DiT block by adding cross-attention layer after the 3D attention layer. In this cross-attention mechanism, the view tokens serve as queries, while the fusion tokens act as keys and values:

$$t^{view} = CrossAttention(t^{view}, t^{fusion}). \tag{4}$$

This allows the rich 2D and 3D information from the reference fusion tokens to be effectively injected into the view tokens, enabling direct alignment with the reference views and enhancing both semantic and geometric consistency when restoring artifact-prone novel views to clean ones.

### 3.2 REFERENCE-GUIDED GENERATIVE RECONSTRUCTION

Given rendered artifact-prone novel views and reference views, our trained reference-guided restoration diffusion model can produce consistent, artifact-free frames aligned with the input sparse views. Leveraging this capability, we supervise 3DGS optimization using both the input sparse views and the fixed novel views in an iterative training manner Haque et al. (2023); Wu et al. (2025b;a). Specifically, we begin by constructing an initial low-quality 3DGS representation from the sparse views. We then render novel views along new camera trajectories and feed these artifact-prone frames into our reference-guided video restoration model to obtain artifact-free outputs, as illustrated in Fig. 2. These fixed novel views are subsequently added to the training set to further supervise the 3DGS optimization iteratively.

**Reference-guided Trajectory.** Trajectory sampling is critical in this iterative optimization process. Common sampling strategies, such as interpolation trajectory between input poses (Fig. 3 (a)) or ellipse trajectory along a spherical path across all the camera poses (Fig. 3 (b)) , have limitations. For our GSFixer, the interpolation trajectory yields high-quality fixed novel views but lacks angular

Figure 4: **Qualitative comparison on DL3DV-Res Benchmark**. We compare 3DGS artifact restoration quality of the existing generative methods.

| | PSNR ↑ | SSIM ↑ | LPIPS ↓ | I2V$_{SC}$ ↑ | I2V$_{BC}$ ↑ | OC ↑ | TF ↑ | MS ↑ |
|---|---|---|---|---|---|---|---|---|
| Artifact (Kerbl et al., 2023) | 14.12 | 0.405 | 0.509 | 0.9382 | 0.9524 | 0.2066 | 0.9058 | 0.9548 |
| Difix3D+ (Wu et al., 2025a) | 14.14 | 0.419 | 0.455 | 0.9288 | 0.9481 | 0.2393 | 0.9038 | 0.9473 |
| GenFusion (Wu et al., 2025b) | 14.56 | 0.453 | 0.486 | 0.8916 | 0.9258 | 0.2372 | 0.9135 | 0.9596 |
| GSFixer (Ours) | **16.72** | **0.520** | **0.399** | **0.9553** | **0.9644** | **0.2407** | **0.9233** | **0.9665** |

Table 1: **Quantitative comparison on DL3DV-Res Benchmark.** We compare the video 3D artifact restoration results with baselines. The best results are highlighted in **bold**.

diversity, while the ellipse trajectory provides broader view coverage but results in suboptimal fixed views. To balance these trade-offs, we propose a reference-guided trajectory sampling strategy, as shown in Fig. 3 (c). Specifically, (i) we first interpolate from a reference view to its nearest viewpoint on the spherical path, (ii) sample additional views along the sphere path, (iii) and then interpolate to the next nearest reference view. This hybrid sampling strategy achieves high-quality fixed views and angle coverage, leading to better 3DGS representation.

During optimization, we freeze the video diffusion model and supervise the 3DGS representation using a loss function $\mathcal{L}$ composed of two components: the reconstruction loss $\mathcal{L}_{recon}$ between rendered images and the input sparse views, and the generative loss $\mathcal{L}_{gen}$ between the rendered novel views and the corresponding fixed novel views. Specifically, we adopt simple photometric losses:

$$\mathcal{L}_{recon} = \lambda_{l1} \cdot \mathcal{L}_{l1} + \lambda_{SSIM} \cdot \mathcal{L}_{SSIM}. \tag{5}$$

The generative loss $\mathcal{L}_{gen}$ consists of the same components as $\mathcal{L}_{recon}$ but is applied to the rendered and fixed novel views:

$$\mathcal{L} = \mathcal{L}_{recon} + \lambda \cdot \mathcal{L}_{gen}, \tag{6}$$

where we employ an annealing strategy to gradually increase the weight of the generative loss $\lambda$ during the 3DGS training.

## 4 EXPERIMENTS

We first evaluate GSFixer on 3DGS artifact restoration against several generative baselines and show its ability in restoring novel views with superior quality and consistency (Sec. 4.1). We further compare our GSFixer with state-of-the-art methods for sparse-view 3D reconstruction and view synthesis (Sec. 4.1). Finally, we ablate our design in Sec. 4.2. ***Please refer to the appendix for the experimental setup, such as training dataset, evaluation benchmarks (including our proposed DL3DV-Res), evaluation metrics, implementation details, and the comparison baselines.***

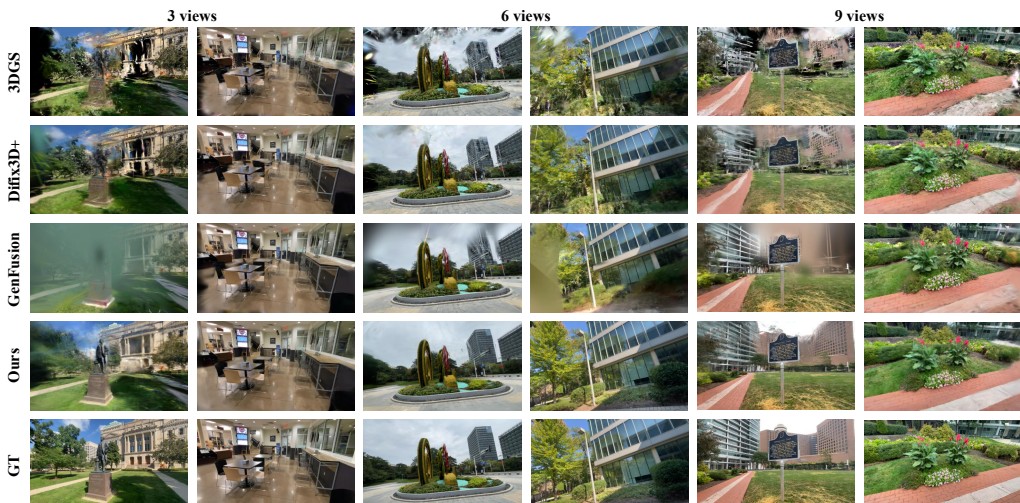

Figure 5: **Qualitative comparison on DL3DV-Benchmark.** We compare the novel view with baselines rendering quality using 3, 6, and 9 input views.

| | PSNR ↑ | | | SSIM ↑ | | | LPIPS ↓ | | |
|---|---|---|---|---|---|---|---|---|---|
| | 3-view | 6-view | 9-view | 3-view | 6-view | 9-view | 3-view | 6-view | 9-view |
| 3DGS (Kerbl et al., 2023) | 13.72 | 17.11 | 19.05 | 0.410 | 0.547 | 0.625 | 0.521 | 0.372 | 0.293 |
| Difix3D+ (Wu et al., 2025a) | 15.07 | 18.26 | 20.12 | 0.481 | 0.589 | 0.656 | 0.473 | 0.329 | 0.259 |
| GenFusion (Wu et al., 2025b) | 14.64 | 18.36 | 20.32 | 0.498 | 0.610 | 0.688 | 0.493 | 0.374 | 0.314 |
| GSFixer (Ours) | 16.21 | 19.11 | 20.60 | 0.536 | 0.625 | 0.675 | 0.478 | 0.360 | 0.307 |

Table 2: **Quantitative comparison about sparse-view 3D reconstruction on DL3DV-Benchmark Dataset.** We compare the rendering quality with baselines given 3, 6 and 9 views.

## 4.1 COMPARISON WITH OTHER METHODS

**3DGS Artifact Restoration.** To evaluate the effectiveness of existing models in 3DGS artifact restoration, we compare our GSFixer with recent generative methods, including Difix3D+ (Wu et al., 2025a) and GenFusion (Wu et al., 2025b) on our proposed DL3DV-Res benchmark. As shown in Tab. 1, the quantitative results demonstrate that GSFixer significantly outperforms both Difix3D+ and GenFusion in correcting artifacts in novel views across all pixel-wise metrics. Specifically, GS-Fixer achieves improvements of 2.16 in PSNR, 0.067 in SSIM, and 0.087 in LIPIS compared to GenFusion. Furthermore, our method consistently outperforms the baselines across all VBench and VBench++ metrics. It is noteworthy that the $I2V_{SC}$ and $I2V_{BC}$ scores of Difix3D+ and GenFusion are even lower than the original artifact frames after their artifact correction, suggesting that these methods struggle to maintain consistency when fixing artifacts in novel views. The qualitative comparisons in Fig. 4 further emphasize GSFixer's superior quality and consistency in fixing the artifacts of novel views. For instance, Difix3D+ and GenFusion fail to restore the content consistently, such as the foreground statue and background buildings.

**In Domain Sparse View Reconstruction.** We next compare our method with baseline approaches for sparse-view 3D reconstruction on the DL3DV-Benchmark (Ling et al., 2024). As shown in Tab. 2, our GSFixer achieves more realistic novel view synthesis across 3, 6, and 9 input views. For example, under the extremely sparse 3-view setting, GSFixer significantly improves 3DGS quality by 3.55 dB in PSNR, 0.119 in SSIM, and 0.034 in LPIPS. Moreover, GSFixer outperforms state-of-the-art methods such as Difix3D+ and GenFusion by a substantial margin. As illustrated in Fig. 5, Difix3D+ fails to generate plausible content in occluded or missing regions, such as the building structure in the first row of the figure. It also does not adequately fix geometry-distorted Gaussians, as seen in the road reconstruction in the fifth and sixth rows. Additionally, inconsistent novel views generated by Difix3D+'s image diffusion process degrade overall quality, particularly noticeable in the statue (first row) and in the building and sky regions (third row). Similarly, GenFusion exhibits notable artifacts in fixing 3DGS representation. For example, its video diffusion process generates "foggy" geometry in the background (statue in first row and building in fifth row), failing to produce 3D-consistent content in missing regions. This results in blurred renderings, as shown in the third and fourth rows. In contrast, our GSFixer effectively inpaints plausible content in missing regions

| | PSNR ↑ | | | SSIM ↑ | | | LPIPS ↓ | | |
|---|---|---|---|---|---|---|---|---|---|
| | 3-view | 6-view | 9-view | 3-view | 6-view | 9-view | 3-view | 6-view | 9-view |
| Zip-NeRF[†] (Barron et al., 2023) | 12.77 | 13.61 | 14.30 | 0.271 | 0.284 | 0.312 | 0.705 | 0.663 | 0.633 |
| FreeNeRF[†] (Yang et al., 2023) | 12.87 | 13.35 | 14.59 | 0.260 | 0.283 | 0.319 | 0.715 | 0.717 | 0.695 |
| SimpleNeRF[†] (Somraj et al., 2023) | 13.27 | 13.67 | 15.15 | 0.283 | 0.312 | 0.354 | 0.741 | 0.721 | 0.676 |
| ZeroNVS[†] (Sargent et al., 2024) | 14.44 | 15.51 | 15.99 | 0.316 | 0.337 | 0.350 | 0.680 | 0.663 | 0.655 |
| ReconFusion[†] (Wu et al., 2024) | 15.50 | 16.93 | 18.19 | 0.358 | 0.401 | 0.432 | 0.585 | 0.544 | 0.511 |
| 3DGS[†] (Kerbl et al., 2023) | 13.06 | 14.96 | 16.79 | 0.251 | 0.355 | 0.447 | 0.576 | 0.505 | 0.446 |
| 2DGS[†] (Huang et al., 2024a) | 13.07 | 15.02 | 16.67 | 0.243 | 0.338 | 0.423 | 0.580 | 0.506 | 0.449 |
| FSGS[†] (Zhu et al., 2024) | 14.17 | 16.12 | 17.94 | 0.318 | 0.415 | 0.492 | 0.578 | 0.517 | 0.468 |
| Difix3D+[‡] (Wu et al., 2025a) | 13.92 | 15.94 | 17.54 | 0.298 | 0.382 | 0.452 | 0.578 | 0.468 | 0.391 |
| GenFusion[‡] (Wu et al., 2025b) | 15.03 | 16.90 | 18.29 | 0.357 | 0.430 | 0.489 | 0.578 | 0.494 | 0.440 |
| GSFixer (Ours) | 15.61 | 17.27 | 18.63 | 0.370 | 0.426 | 0.481 | 0.559 | 0.478 | 0.420 |

Table 3: **Quantitative Comparison on Mip-NeRF 360 Dataset.** We compare the rendering quality with baselines given 3, 6 and 9 views. † denotes results reproduced by ReconFusion and GenFusion, while ‡ indicates results reproduced by us on their official implementation.

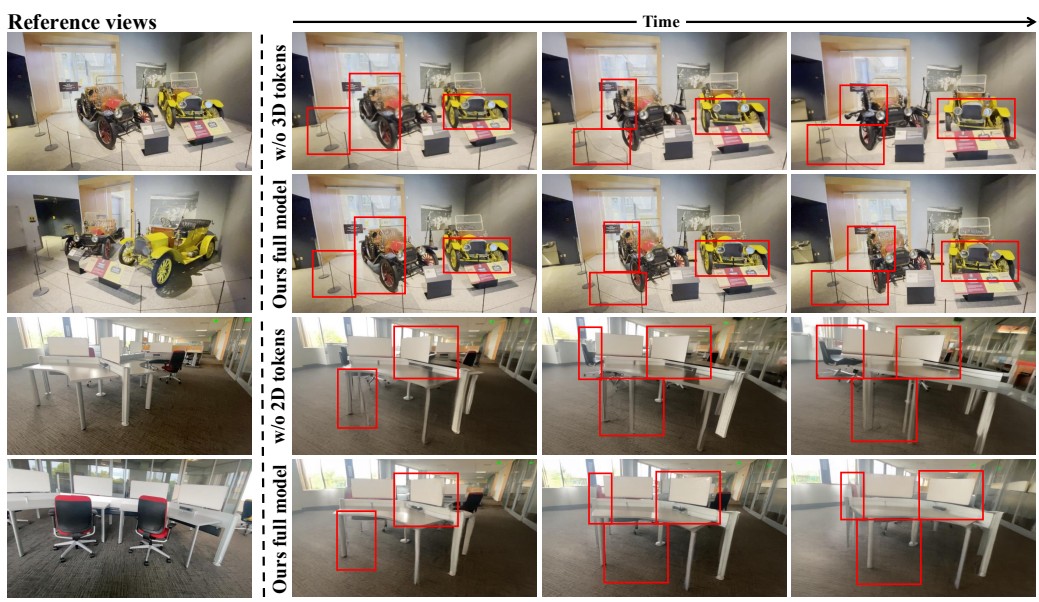

Figure 6: **Effectiveness of reference view conditions.** We compare our full model with two alternatives: a variant without 3D conditions (top), and a variant without 2D conditions (bottom). The red boxes highlight the most prominent differences.

and performs both semantically and geometrically consistent restoration in novel views, leading to high-quality and coherent novel view renderings.

**Out of Domain Sparse View Reconstruction.** To demonstrate the generalizability of our method for out-of-distribution dataset, we further evaluate the methods on the challenging Mip-NeRF 360 dataset (Barron et al., 2022) with 3, 6, and 9 input views. The quantitative results presented in Tab. 5 show that our GSFixer outperforms the baseline approaches, highlighting its strong generalization performance on unseen and complex scenes. For instance, our GSFixer surpasses recent generative 3DGS-based methods such as Difix3D+ and GenFusion. Moreover, we observe that our GSFixer even outperforms generative NeRF-based ReconFusion in sparse view settings. This is particularly notable as 3DGS is generally more prone to overfitting on sparse input views compared to NeRF. *For more visualization results, please refer to the appendix.*

## 4.2 ABLATION STUDIES

**Effectiveness of Reference View Conditions.** We conduct an ablation study on one of our key contributions: the reference-guided video restoration model, which controls the 3DGS artifact restoration process using 3D and 2D tokens extracted from reference views. We compare the video restoration results of three variants on the DL3DV-Res benchmark: our full model, a version without 3D tokens, and one without 2D tokens. As quantitatively results presented in Tab. 4, our full model consistently outperforms the other two variants across all metrics. The qualitative comparisons in

| | PSNR ↑ | SSIM ↑ | LPIPS ↓ | I2V$_{SC}$ ↑ | I2V$_{BC}$ ↑ | OC ↑ | TF ↑ | MS ↑ |
|---|---|---|---|---|---|---|---|---|
| Ours w/o 3D tokens | 16.36 | 0.510 | 0.414 | 0.9527 | 0.9630 | 0.2403 | 0.9218 | 0.9663 |
| Ours w/o 2D tokens | 16.48 | 0.516 | 0.409 | 0.9540 | 0.9635 | 0.2405 | 0.9225 | 0.9665 |
| Ours full model | **16.72** | **0.520** | **0.399** | **0.9553** | **0.9644** | **0.2407** | **0.9233** | **0.9665** |

Table 4: **Ablation study about different conditions.** We report the PSNR, SSIM, LPIPS, VBench and VBench++ metrics for the full model and its ablated versions on DL3DV-Res.

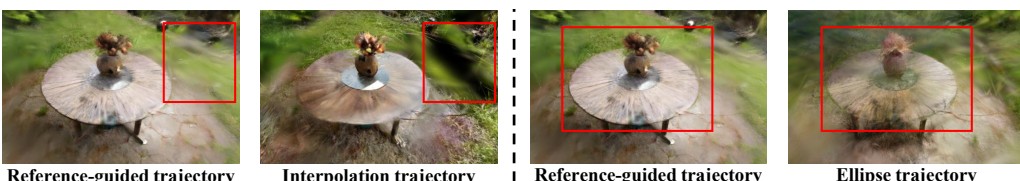

**Reference-guided trajectory**   **Interpolation trajectory**   |   **Reference-guided trajectory**   **Ellipse trajectory**

Figure 7: **Effectiveness of reference-guided trajectory.** We compare with widely used interpolation trajectory and ellipse trajectory. The red boxes highlight the most prominent differences.

| | PSNR ↑ | SSIM ↑ | LPIPS ↓ |
|---|---|---|---|
| Interpolation trajectory | 14.50 | 0.353 | 0.565 |
| Ellipse trajectory | 15.46 | 0.362 | 0.563 |
| Reference-guided trajectory | **15.61** | **0.370** | **0.559** |

Table 5: **Ablation study about trajectory sampling.** We report the PSNR, SSIM, and LPIPS metrics for different trajetories on Mip-NeRF 360.

Fig. 6 further support this finding. From the visualization, we observe that removing the 3D condition leads to poor 3D consistency with the reference views, e.g., misalignment in the fence and cars (highlighted in red boxes). This is because 3DGS renders under sparse-view settings often exhibit severe geometric distortions, making it difficult for the diffusion model to generate geometrically consistent content without strong 3D priors. Similarly, without the 2D condition, the model fails to restore semantically plausible details, such as the table leg, baffle, socket, and chair marked in red boxes. In contrast, our full model successfully restores artifact-ridden novel views to frames that are both 3D consistent and semantically aligned with the reference views, demonstrating the effectiveness of injecting both 3D and 2D conditions into the video diffusion model.

**Effectiveness of Reference Guided Trajectory.** We further conduct an ablation study about different trajectory sampling on the Mip-NeRF 360 dataset using a 3-view reconstruction setting. Qualitative results are presented in Fig. 7, the interpolation trajectory lacks sufficient angular coverage, leading to incomplete reconstructions with missing regions. Meanwhile, the ellipse trajectory results in degraded rendering quality, as the novel views to be restored are not temporally sequential with the reference views. In contrast, our reference-guided trajectory ensures both angular coverage and view quality in the reconstruction process. The quantitative results in Tab. 5 further confirm the effectiveness of the reference-guided trajectory sampling strategy.

## 5 CONCLUSIONS

We present GSFixer, a novel pipeline for enhancing the quality of 3D Gaussian Splatting in sparse-view 3D reconstruction and novel view synthesis. Specifically, our reference-guided video restoration model integrates both 3D geometric and 2D semantic condition signals from reference views, ensuring semantically coherent and geometrically consistent artifact correction in novel views. Furthermore, we introduce a reference-guided trajectory sampling strategy within the iterative reconstruction process, enabling high-quality 3D reconstruction. In addition, we propose the DL3DV-Res benchmark to evaluate the 3DGS artifact restoration capability.

**Limitations and Future Work:** Our GSFixer builds upon a DiT-based video diffusion model that requires 50 denoising steps, which may limit the efficiency. As a 3D enhancement model, its performance is inherently limited by the quality of the initial 3DGS representation. Future work may explore improving GSFixer with advanced single-step video diffusion models and improved 3D representation, enabling efficient and high-fidelity novel view synthesis.

## 6 ETHICS STATEMENT

As with all generative AI technologies, our method carries potential risks of misuse, such as generating misleading or synthetic 3D content. We caution against deceptive applications of this technology, particularly in contexts where realistic generative outputs could produce unverifiable or misleading media. To mitigate these risks, we advocate for the adoption of safeguards such as provenance tracking, dataset documentation, and rigorous evaluation practices. These measures will help ensure that generative models contribute positively to scientific progress and societal benefit.

## 7 REPRODUCIBILITY STATEMENT

To facilitate reproducibility, we provide implementation details and per-scene results in both the main paper and the appendix. Furthermore, to enable full replication of our work and support future research, we will release our code, model checkpoints, and benchmark for both 3DGS artifact restoration and sparse-view 3D reconstruction.

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

# Appendix: GSFixer: Improving 3D Gaussian Splatting via Reference-Guided Video Diffusion Priors

## A  Preliminary

**3D Gaussian Splatting and Novel View Synthesis.** 3DGS (Kerbl et al., 2023) explicitly represents a 3D scene as a collection of 3D Gaussian spheres, enabling high-quality 3d reconstructions and efficient novel view synthesis. Each 3D Gaussian sphere is defined by its center location $\mu$, scaling vector $s$, rotation quaternion $q$, opacity $\sigma$, and spherical harmonic (SH) coefficients $sh$. Thus, the Gaussian distribution is formulated as:

$$G(x) = e^{-\frac{1}{2}(x-\mu)^T \Sigma^{-1} (x-\mu)}, \tag{7}$$

where $\Sigma = RSS^T R^T$, $S$ denotes the scaling matrix corresponding to $s$ and $R$ is the rotation matrix determined by $q$. For rendering novel views, volume rendering integrates these elements using:

$$C = \sum_{i \in M} c_i \alpha_i \prod_{j=1}^{i-1} (1 - \alpha_j), \tag{8}$$

where $C$ represents the final pixel color, which is computed via alpha blending of the view-dependent colors $c_i$ of the $M$ contributing Gaussians, weighted by their opacities $\alpha_i$..

**Video Diffusion Model.** Video diffusion models (Blattmann et al., 2023; Chen et al., 2023; 2024; Xing et al., 2024) typically consist of two key stages: a forward diffusion process, which progressively inject noise $\epsilon$ into clean video data $\boldsymbol{x}_0 \in \mathbb{R}^{n \times 3 \times h \times w}$, yielding noisy samples $\boldsymbol{x}_t = \alpha_t \boldsymbol{x}_0 + \sigma_t \epsilon$ at each time step $t$; and a reverse denoising process $p_\theta$, where a noise predictor $\epsilon\theta$ learns to recover the original data by removing the noise. This predictor was traditionally implemented using a U-Net architecture and is trained to minimize the following denoising objective:

$$\min_\theta \mathbb{E}_{t \sim \mathcal{U}(0,1), \epsilon \sim \mathcal{N}(\boldsymbol{0}, \boldsymbol{I})} [\|\epsilon_\theta(\boldsymbol{x}_t, t) - \epsilon\|_2^2]. \tag{9}$$

Following Sora (Brooks et al., 2024), recent approaches (Yang et al., 2024; Wan et al., 2025) adopt the Diffusion Transformer (DiT) (Peebles & Xie, 2023) architecture for the noise predictor. During training, a pretrained 3D VAE encoder $\mathcal{E}$ compresses videos into latent space $\boldsymbol{z} = \mathcal{E}(\boldsymbol{x})$. These latent tokens $\boldsymbol{z}$ are then patchified, concatenated with text tokens, and fed into the DiT. At inference time, the model progressively denoises the latent tokens into clean tokens, which are subsequently decoded by the 3D VAE decoder $\mathcal{D}$ to generate the final video $\hat{\boldsymbol{x}} = \mathcal{D}(\boldsymbol{z})$.

## B  Reference-Guided Video Diffusion Model Details

Our reference-guided video diffusion model is built upon CogVideoX-5B-I2V, a pretrained DiT-based video diffusion model capable of generating 49-frame videos at a resolution $480 \times 720$ from a single input image. In our video-to-video restoration task, which aims to restore artifact frames into

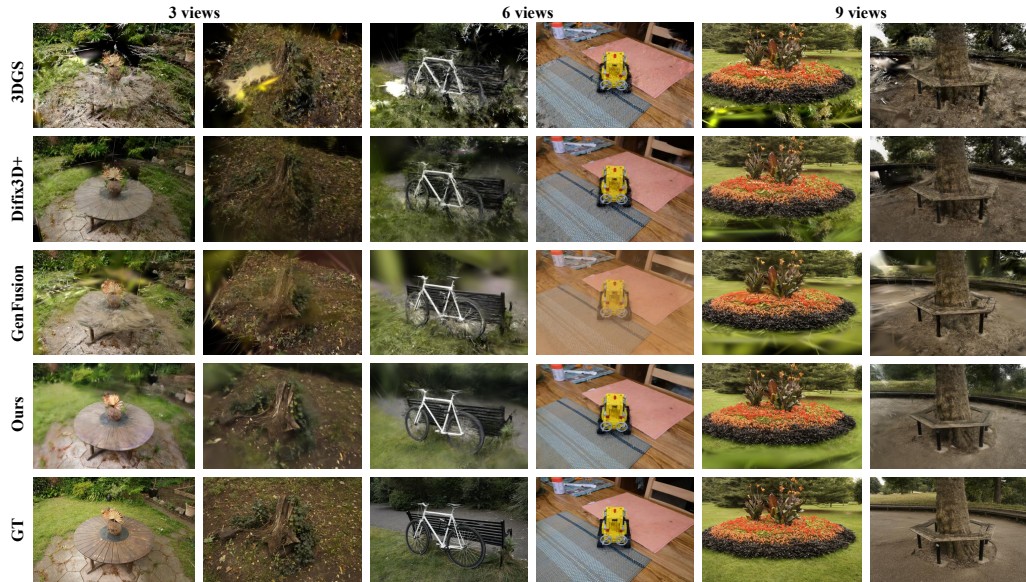

Figure 8: **Qualitative comparison on Mip-NeRF 360.** We compare the novel view rendering quality with baselines using 3, 6, and 9 input views.

clean ones, we replace the original image condition with the 3DGS renders $\{I_n^{nov}\}_{n=1,...,47}$ between two reference views $\{I_1, I_2\}$. In the training process, each training sample consists of a sequence of 47-frame artifact-prone 3DGS renders, two reference images, and 49-frame ground truth RGB video. The artifact frames ($49\times3\times480\times720$) and ground truth frames ($49\times3\times480\times720$) are first encoded into latent features using a 3DVAE, resulting in latent dimensions of $16\times13\times60\times90$. The ground truth latent features are then perturbed with noise for the diffusion training process. For the view tokens, we concatenate, patchify and project the latent features to $17550\times3072$ view tokens. For the text tokens, we use BLIP to generate text captions of the reference views and use T5 to extract $226\times4096$ feature embedding, which are projected to $226\times3072$ text tokens. To extract geometric and semantic conditioning signals, the two reference views are first resize to $350\times518$ to feed into VGGT and DINOv2 encoder to produce $2\times930\times2048$ geometric tokens and $2\times930\times1024$ semantic tokens. We abondon $2\times5\times2048$ camera token from VGGT, and remove $2\times1\times1024$ class token and $2\times4\times1024$ registration token from DINOv2, resulting in $2\times925\times2048$ 3D geometric tokens and $2\times925\times1024$ 2D semantic tokens. These are projected to $2\times925\times3072$ using $Projector_{3D}$ and $Projector_{2D}$, respectively. $Projector_{3D}$ is a linear layer mapping from 2048 to 3072 dimensions, followed by LayerNorm. $Projector_{2D}$ is a linear layer mapping from 1024 to 3072 dimensions, followed by LayerNorm. Finally, the 3D geometric and 2D semantic tokens are combined and reshaped into fusion tokens of dimension $1850\times3072$, which serve as rich 3D and 2D priors to guide the restoration process of diffusion model. During inference, we employ DDIM sampling with classifier-free guidance to modulate condition adherence strength.

## C  DETAILS OF EXPERIMENTAL SETUP

**Training Dataset and Evaluation Benchmarks.** We train our GSFixer on a random selection of 1,000 scenes from the DL3DV-10K dataset (Ling et al., 2024). To construct paired artifact-laden and clean 3DGS renders, we employ a sparse-view 3D reconstruction strategy. Specifically, we use a few of input views (e.g., 3, 6, or 9) to reconstruct low-quality 3DGS representations. We then select camera trajectories from the dataset that include both the input views (used as reference views) and novel viewpoints (serving as ground truth frames). By rendering the low-quality 3DGS along these trajectories, we can generate the corresponding artifact-containing frames. Additionally, we extract the 1D text captions, 2D semantic tokens and 3D geometric tokens of the reference views by the pretrained BLIP, DINOv2 and VGGT encoders, respectively.

| Method | Restoration | | Reconstruction | | |
|---|---|---|---|---|---|
| | Time (s/frame) | Memory (GB) | Time (min/scene) | Memory (GB) | Iterations |
| Difix3D+ | 0.75 | 21.9 | 68 | 25.3 | 29 |
| GenFusion | 1.09 | 19.3 | 39 | 28.2 | 5 |
| GSFixer | 4.32 | 25.6 | 72 | 49.8 | 2 |

Table 6: Computational and memory comparison of restoration and reconstruction.

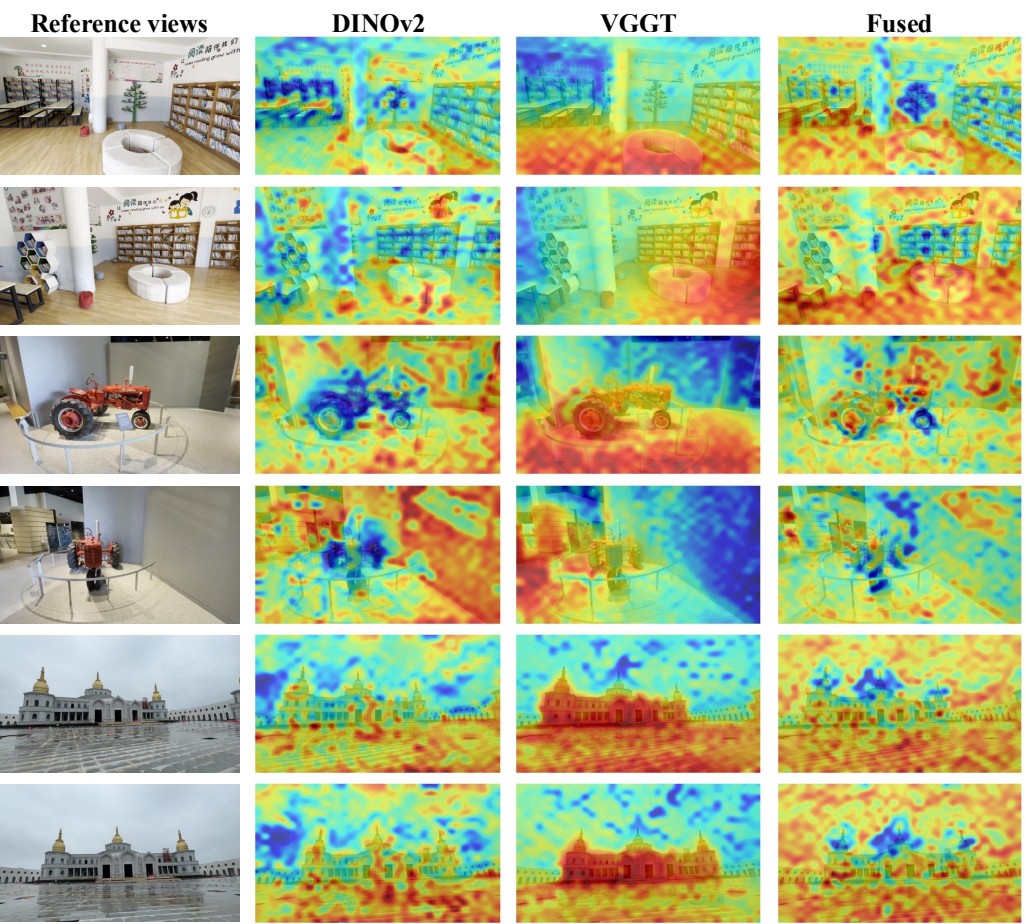

Figure 9: Attention heatmap visualization about different conditioning signals.

We evaluate our method and other baselines on 137 scenes from our proposed DL3DV-Res benchmark for the 3DGS artifact restoration task, and 28 scenes from the DL3DV-Benchmark (Ling et al., 2024) and 9 scenes from the Mip-NeRF 360 dataset (Barron et al., 2022) for the sparse-view 3D reconstruction task. For evaluation, we report PSNR, SSIM, and LPIPS metrics and Overall Consistency (OC), Temporal Flickering (TF), Motion Smoothness (MS) scores of VBench (Huang et al., 2024b) and Image-to-Video Subject Consistency ($I2V_{SC}$), Image-to-Video Background Consistency ($I2V_{BC}$) scores of VBench++ (Huang et al., 2024c) protocol.

The proposed DL3DV-Res benchmark is constructed from all available scenes in the DL3DV-Benchmark, using a similar sparse-view reconstruction protocol. We train 3DGS models with extremely sparse input views (e.g., 3) and render the reconstructions along the original camera trajectories of each scene. This produces novel views with severe artifacts, paired with high-quality ground truth images. For fair comparison, all the methods initialize with the COLMAP (Schonberger & Frahm, 2016) point cloud in all the experiments and we filter and retain only the visible points from the input sparse training views.

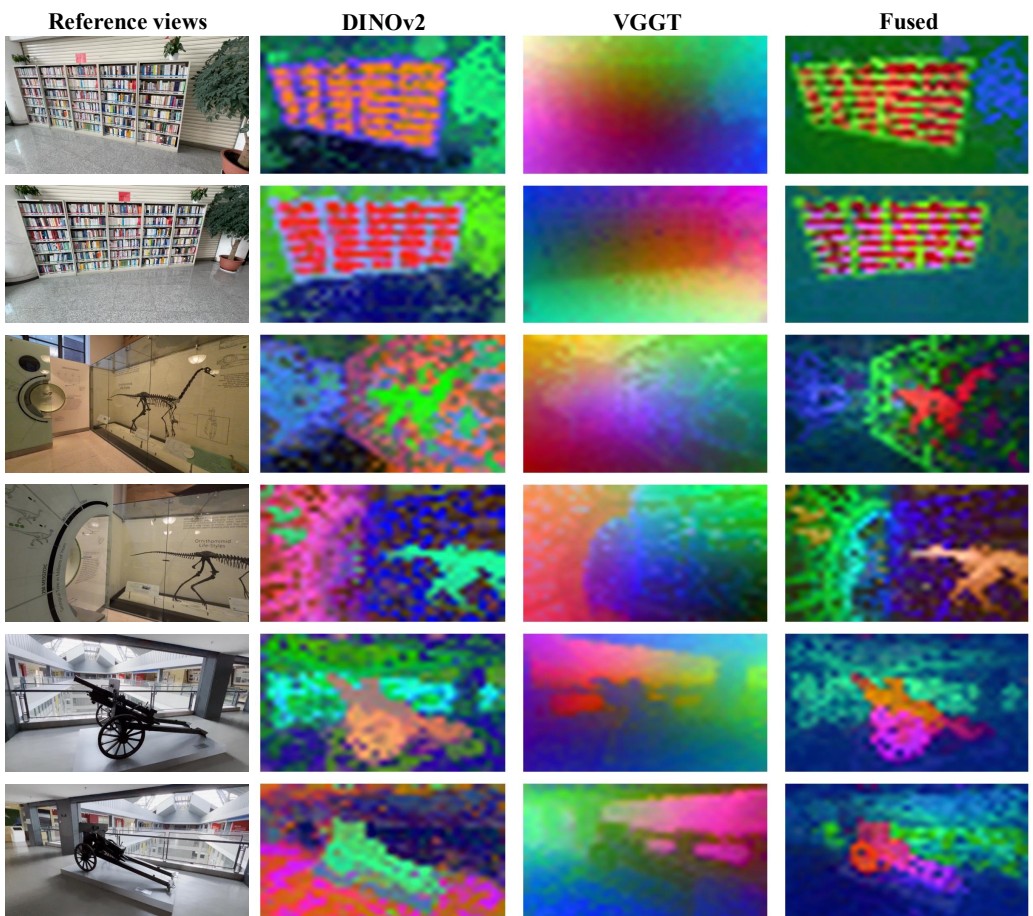

Figure 10: PCA visualization of different conditioning signals.

**Implementation Details.** We implement and initialize the parameters of our reference-guided video diffusion model based on the pretrained CogVideoX-5B-I2V (Yang et al., 2024). During training, the frame resolution is fixed at $480 \times 720$, and the video length is set to 49 frames. The training stage is conducted for 10,000 iterations with a learning rate of $2 \times 10^{-5}$, incorporating a warm-up strategy, and optimized using the AdamW optimizer. The proposed model is trained on 8 NVIDIA H20 GPUs with a batch size of 8 for about 4 days.

**Comparison Baselines.** We compare our method with (i) regularization methods and (ii) generative methods for sparse-view reconstruction and 3DGS artifact restoration tasks. Specifically, we evaluate NeRF-based per-scene regularization methods including Zip-NeRF Barron et al. (2023), FreeN-eRF Yang et al. (2023) and SimpleNeRF Somraj et al. (2023); NeRF-based generative methods such as ZeroNVS Sargent et al. (2024) and ReconFusion Wu et al. (2024); 3DGS-based per-scene regularization method FSGS Zhu et al. (2024); and 3DGS-based generative methods Difix3D+ Wu et al. (2025a) and GenFusion Wu et al. (2025b). Difix3D+ incoporates generative priors from the image diffusion model SD-turbo Sauer et al. (2024), finetuned on paired artifact renders and clean frames. In contrast, GenFusion bulids upon the Unet-based video diffusion model DynamicCrafter Xing et al. (2024) that condition video frames on artifact-prone RGBD renders.

## D COMPUTATIONAL AND MEMORY COST ANALYSIS

We provide a detailed breakdown of runtime and memory usage for GSFixer and key baselines (Difix3D+ and GenFusion), including both the restoration stage and the full reconstruction pipeline. All results in the Tab. 6 are averaged over the DL3DV-Res benchmark (restoration) and Mip-NeRF 360 (3, 6, and 9-view reconstruction), evaluated on a single NVIDIA H100 GPU. From the re-

Figure 11: **Qualitative comparison on AerialMegaDepth dataset.** We compare the novel view rendering quality using 9, 12, and 24 input views.

| | PSNR ↑ | | | SSIM ↑ | | | LPIPS ↓ | | |
| --- | --- | --- | --- | --- | --- | --- | --- | --- | --- |
| | 9-view | 12-view | 24-view | 9-view | 12-view | 24-view | 9-view | 12-view | 24-view |
| GenFusion | 11.23 | 11.43 | 13.07 | 0.399 | 0.411 | 0.477 | 0.578 | 0.562 | 0.520 |
| GSFixer (ellipse) | 11.99 | 12.57 | 13.56 | 0.406 | 0.433 | 0.495 | 0.570 | 0.553 | 0.499 |
| GSFixer (reference) | **12.34** | **12.96** | **14.01** | **0.424** | **0.459** | **0.515** | **0.561** | **0.533** | **0.489** |

Table 7: **Quantitative comparison on AerialMegaDepth dataset.** We compare the novel view rendering quality using 9, 12, and 24 input views.

sults we can observe that while our per-frame restoration time (4.32s) is higher due to the 50-step CogVideoX-I2V-5B based model, our total scene reconstruction time remains comparable to Difix3D+. This is because Difix3D+ utilizes a fast one-step image diffusion model (0.75s/frame) but requires 29 fix-and-distill iteration to achieve the final results, while our GSFixer use two cycles in all experiments. Our work focuses on solving the inconsistency issues to improve the performance, and future work (e.g., caching (Kahatapitiya et al., 2025), few-step distillation (Huang et al., 2025)) can accelerate GSFixer without modifying the pipeline. Although our GSfixer use the advanced video diffusion model (CogVideoX-I2V-5B) and extra encoders (VGGT, DINOv2, BLIP), which needs higher memory. However, this cost is justified by the significant improvements in fidelity and consistency (e.g., +2.16 dB PSNR over GenFusion and Difix3D+ on DL3DV-Res). We also note that the recent video diffusion model (Wan-1.3B (Wan et al., 2025)) achieves comparable generation quality with much lower memory requirements and can be adopted in future extensions of GSFixer without modifying the pipeline.

# E MORE VISUALIZATION AND ANALYSIS OF CONDITIONING SIGNALS

Besides the quantitative and qualitative ablation studies described in Tab. 4 and Fig. 6, we further provide some theoretical analysis and visualization of conditioning features to better understand different conditioning signals.

Our fusion implementation is designed to handle the potential issue of scales, dimensions, and information densities of the geometric features (VGGT) and semantic features (DINOv2): 1) Normalization: the geometric and semantic features are processed separately through MLPs projectors (linear and normalization layers). The normalization layers ensure that both geometric and semantic features are brought to the same magnitude scale. 2) Dimension Mapping: the linear layers map the dimensions of both features to match the dimension of the diffusion model's intermediate layer. 3) Adaptive Weighting: after passing through the projectors, the information density of the geometric and semantic features, will be adaptively weighted after our fine-tuning process.

We provide attention heatmap visualizations of different condition signals in Fig. 9, which illustrates how different features prioritize information in the reference images, leading to distinct patterns for DINOv2, VGGT, and the Fused features. From the visualization, we can observe that the VGGT

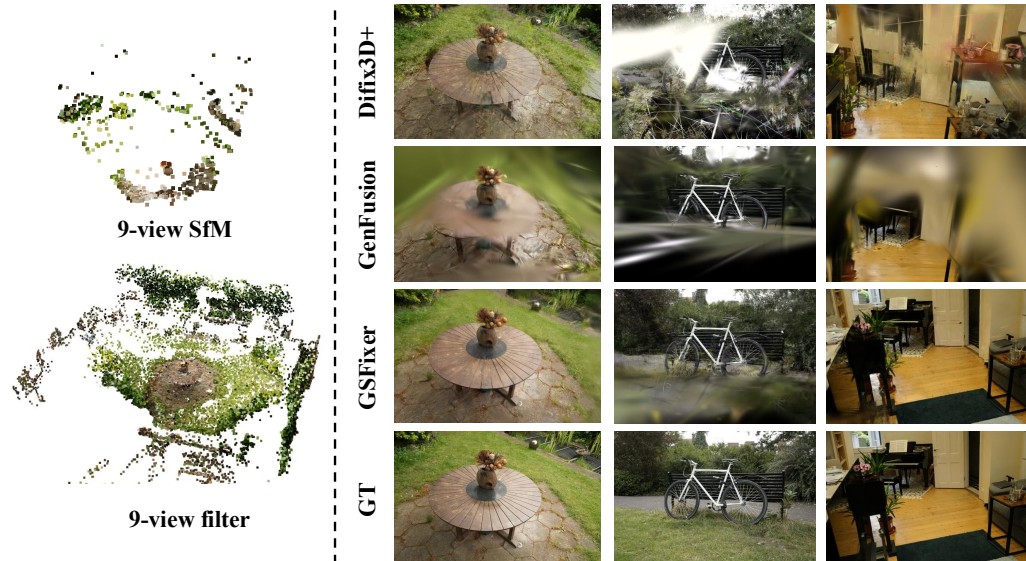

Figure 12: Qualitative comparison with sparse-only SfM point cloud initialization on Mip-NeRF dataset under 9-view setting.

| Method | PSNR ↑ | SSIM ↑ | LPIPS ↓ |
|---|---|---|---|
| 3DGS | 13.82 | 0.287 | 0.559 |
| Difix3D+ | 14.60 | 0.315 | 0.500 |
| GenFusion | 14.71 | 0.362 | 0.643 |
| GSFixer (Ours) | **16.57** | **0.383** | **0.503** |

Table 8: Quantitative comparison sparse-only SfM point cloud initialization on Mip-NeRF 360 dataset using 9-view setting.

attention heatmap shows it activation primarily concentrated along sharp geometric details, such as edges, corners, boundaries, while the DINOv2 attention heatmap exhibits high activation is broad and continuous across the entire area of recognized objects. The Fused feature map combines the strengths of both VGGT and DINOv2: high activation over the entire semantic area (like DINOv2), but with the activation map being enhanced and refined by geometric boundaries (like VGGT). This indicates the fusion successfully captures a holistic representation that is robust to global context while retaining local geometry awareness.

In addition, we further provide Principal Component Analysis (PCA) visualization in Fig. 10. From the visualization, we can observe that PCA visualization of DINOv2 features shows large, contiguous blocks of color that align with foreground objects or background regions, demonstrating its ability to capture semantic clusters. The fused features exhibits the desired combination: stable semantic clusters (inherited from DINOv2) whose boundaries are now refined with the detail geometry (inherited from VGGT).

# F MORE RESULTS ON COMPLEX SCENES

We conduct experiments on six scenes of AerialMegaDepth dataset (Vuong et al., 2025), which features aerial photography scenes captured with discontinuous trajectories and extreme viewing angles. These aerial scenes are unseen in the training data with unseen extreme viewpoint, and irregular captured trajectories, which are significantly more challenging than standard object-centric scenes of Mip-NeRF 360 dataset. We compare our GSFixer with reference-guided trajectory and ellipse trajectory with GenFusion. The results in Tab. 7 demonstrate that GSFixer with the proposed reference-guided trajectory remains robust under diverse distribution of reference viewpoints.

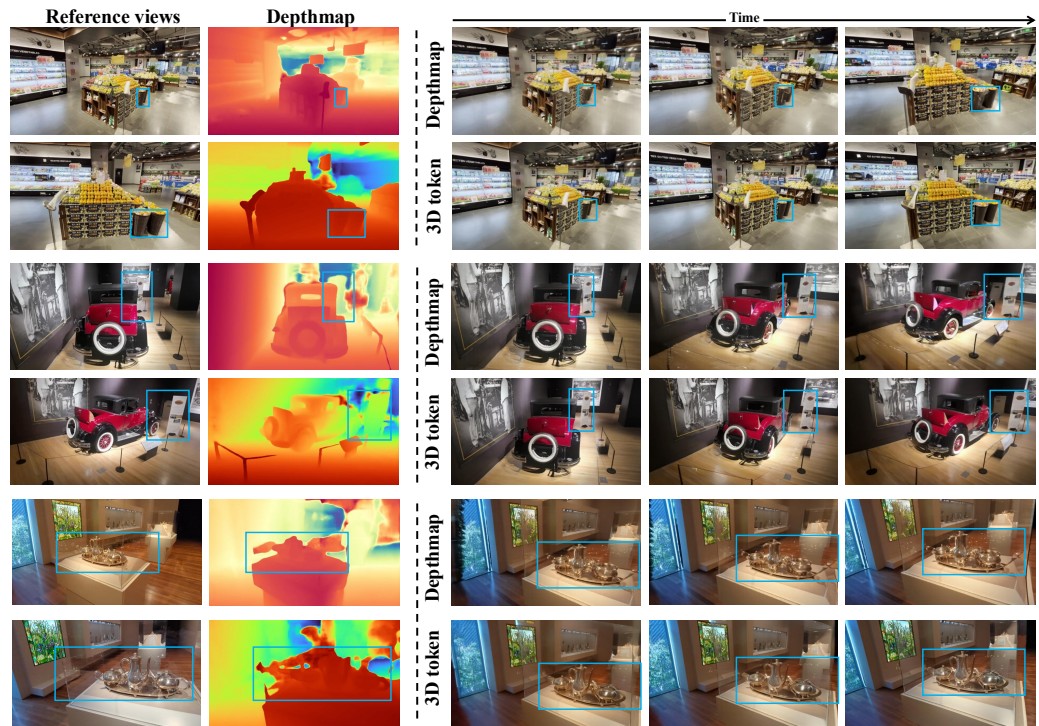

Figure 13: **Effectiveness of using implicit 3D conditioning signal.** We compare using latent features of VGGT with depthmap output of VGGT. The blue boxes highlight some prominent differences.

| Method | PSNR ↑ | SSIM ↑ | LPIPS ↓ |
|---|---|---|---|
| depthmap | 15.13 | 0.485 | 0.431 |
| 3D features | **16.48** | **0.516** | **0.409** |

Table 9: **Ablation study about using implicit or explicit 3D condition signals.** We report the PSNR, SSIM, and LPIPS metrics for using the depthmap output and 3D latent features of VGGT on DL3DV-Res benchmark.

GSFixer consistently outperforms GenFusion in these complex environments further verify the effectivenesss of our method. We also provide some qualitative comparison in Fig. 11.

# G MORE RESULTS WITH SFM INITIALIZATION

We conduct experiments on the Mip-NeRF 360 dataset, where the initial point cloud is generated using only a few images without any evaluation images. For extremely few images, such as 3 views, COLMAP fails to generate points, so we conduct this evaluation using the 9-view setting. The results are presented in Tab. 8. While the use of sparse-only initialization results in a performance drop across all methods due to the lower quality of the initial geometry, GSFixer significantly outperforms the baselines. This demonstrates that our method has stronger generative capability and robustness than the baselines. We also provide some point cloud visualization about initial point clouds that are generated using only a few images (9-view SfM) and an initial point cloud that uses dense COLMAP points and filtering for visibility (9-view filter), and qualitative comparison in Fig. 12.

| Difix3D+ w/o ref | Difix3D+ w one ref | Difix3D+ w two ref | GT |
| --- | --- | --- | --- |

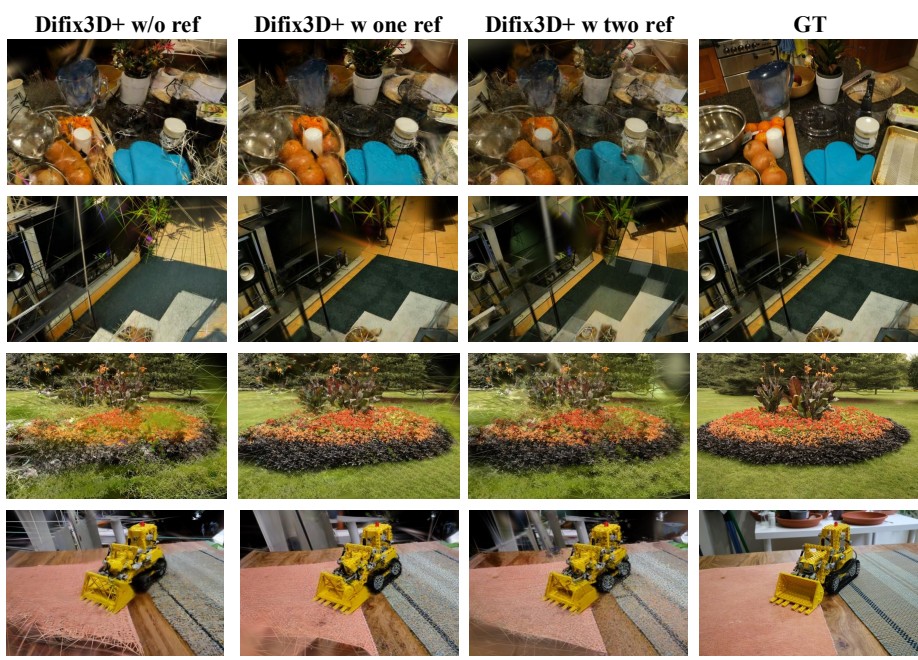

Figure 14: Visualizations of Difix3d+ without, with one nearest and two nearest reference views.

| Method | PSNR ↑ | SSIM ↑ | LPIPS ↓ |
| --- | --- | --- | --- |
| Difix3D+ (no ref) | 13.61 | 0.274 | 0.599 |
| Difix3D+ (one ref) | 13.92 | 0.298 | 0.578 |
| Difix3D+ (two ref) | 13.86 | 0.294 | 0.595 |

Table 10: Quantitative results of Difix3d+ without, with one nearest and two nearest reference views.

## H  MORE ABLATION ABOUT USING IMPLICIT OR EXPLICIT 3D CONDITIONING SIGNALS

We conduct experiments about using implicit latent features or explicit 3D depthmap output of VGGT on DL3DV-Benchmark in Tab. 9. Explicit depth maps generated from sparse views using VGGT may contain noise or structural errors. When used as a direct condition, these errors may propagate into the restoration process, forcing the model to adhere to incorrect geometry. In contrast, conditioning on latent features of VGGT allows the model to leverage rich geometric information. This implicit guidance enables the diffusion model to utilize geometric cues without overfitting to the inaccuracies present in depthmap. We also provide visual comparisons in Fig. 13, which further demonstrate that our implicit feature-based conditioning yields superior robustness compared to explicit depthmap conditioning.

## I  MORE RESULTS AND ANALYSIS ABOUT DIFIX3D+

Although Difix3D+ briefly mentions the possibility of using multiple reference views, its official implementation and results rely on using either no reference or one nearest reference view. It lacks a dedicated mechanism to effectively fuse information from multiple views. To verify this empirically, we extend Difix3D+ to use two nearest reference views on the Mip-NeRF 360 dataset under 3-view setting. As shown in Tab. 10, Difix3D+ with two nearest reference views performance slightly degrades compared to the single-view baseline. This indicates that Difix3D+ cannot be straight-forwardly extended to multi-reference conditioning. In contrast, our GSFixer is explicitly designed for multi-view reference fusion, leveraging both 3D geometry (VGGT) and 2D semantic features (DINOv2) from multiple clean reference views via cross-attention, which effectively addresses the consistency issues where Difix3D+ struggles. We further provide qualitative comparison in Fig. 14.

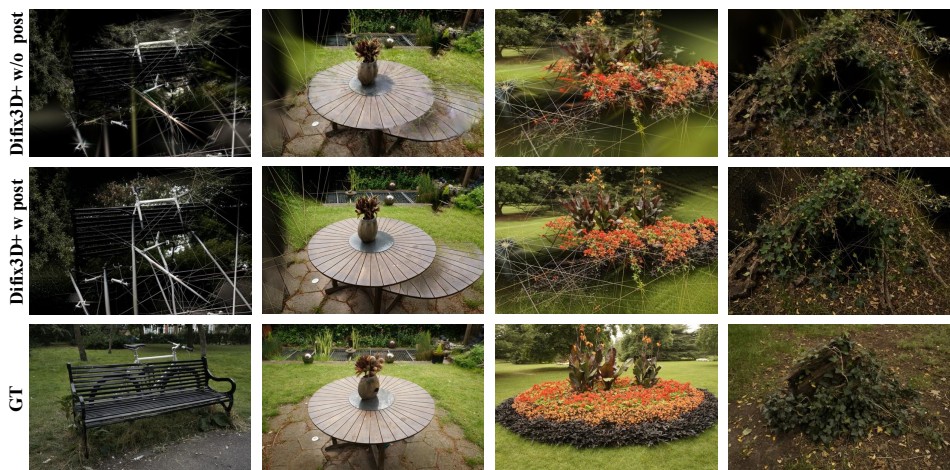

Figure 15: Visualization of Difix3d+ with and without post-rendering refinement step.

| | PSNR ↑ | | | SSIM ↑ | | | LPIPS ↓ | | |
| --- | --- | --- | --- | --- | --- | --- | --- | --- | --- |
| | 3-view | 6-view | 9-view | 3-view | 6-view | 9-view | 3-view | 6-view | 9-view |
| Difix3D+ w/o post | 13.92 | 15.94 | 17.54 | 0.298 | 0.382 | 0.452 | 0.578 | 0.468 | 0.391 |
| Difix3D+ w post | 13.77 | 15.70 | 17.20 | 0.263 | 0.342 | 0.401 | 0.549 | 0.460 | 0.411 |

Table 11: Quantitative results of Difix3d+ with and without post-rendering refinement step.

The results reported for Difix3D+ in our main paper were obtained without the post-rendering refinement step. For completeness, we provide the results of Difix3D+ with and without post-rendering refinement on the Mip-NeRF 360 dataset in the Tab. 11. The refinement step is effective for dense-view settings, which are demonstrated in Difix3D+ paper. However, in the sparse-view settings, the final 3DGS renders contain severe artifacts and large missing regions. As illustrated in Fig. 15, applying the additional post-rendering refinement step tends to amplifies these ambiguous or distorted geometry regions, leading to lower fidelity compared to the variant without the refinement.

## J   MORE QUANTITATIVE AND QUALITATIVE RESULTS ON DL3DV-RES, MIP-NERF 360 AND DL3DV-BENCHMARK

Qualitative results about sparse-view 3D reconstruction on Mip-NeRF 360 are shown in Fig. 8. The visualization further illustrate the semantic and 3D consistent generative capability of our GSFixer. We also provide extensive per-scene experimental results in Fig. 16, Tab. 12, Tab. 13, Tab. 14, Tab. 15, Tab. 16, and Tab. 17.

## K   THE USE OF LARGE LANGUAGE MODELS

We only utilize LLMs to refine the writing style and enhance the clarity of exposition. The LLMs are not involved in research ideation, experimental design or data analysis.

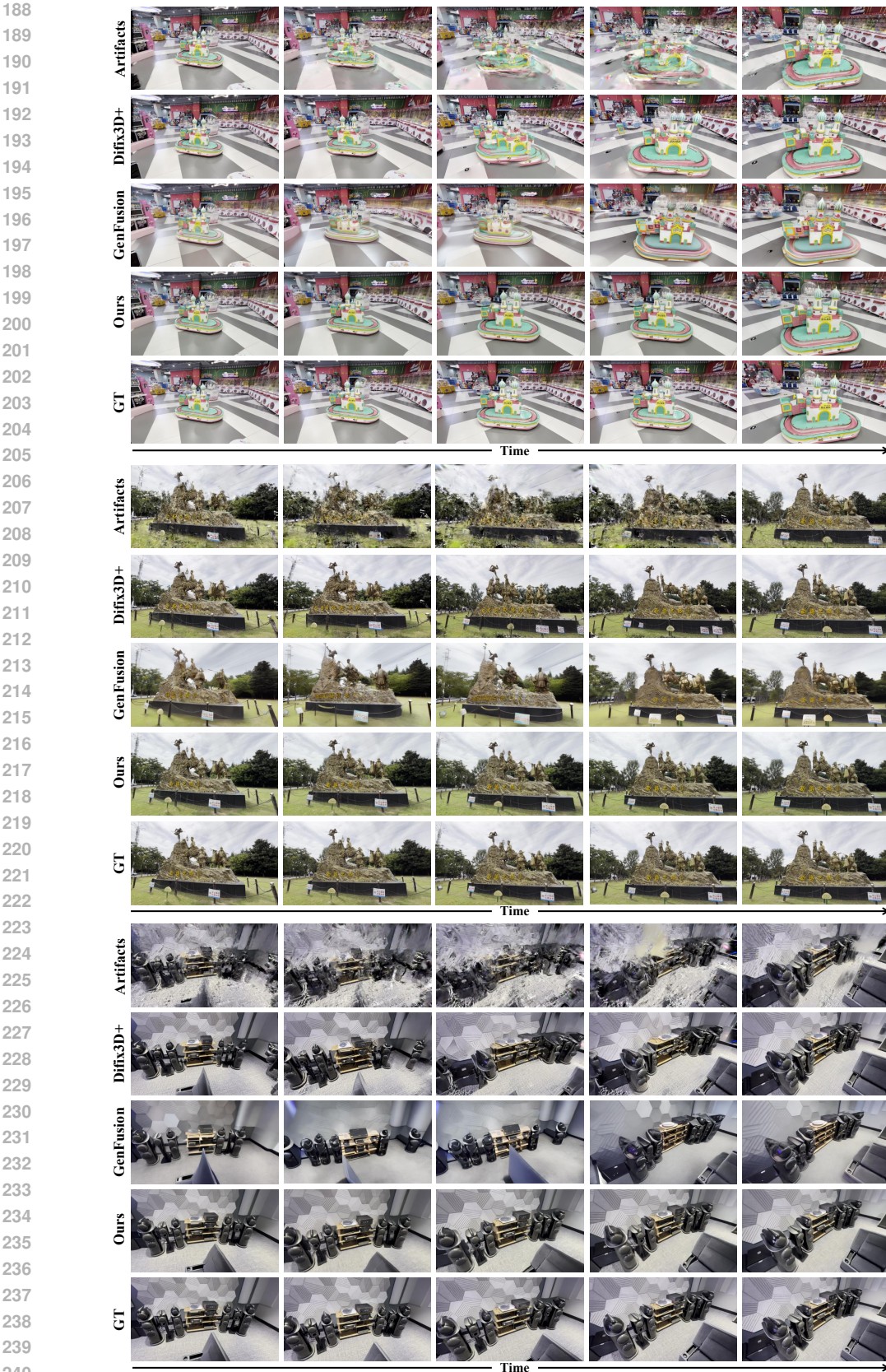

Figure 16: Qualitative comparision of 3DGS artifact restoration on DL3DV-Res.

| | Artifact | | | Difix3D+ | | | GenFusion | | | GSFixer (Ours) | | |
|---|---|---|---|---|---|---|---|---|---|---|---|---|
| | PSNR↑ | SSIM↑ | LPIPS↓ | PSNR↑ | SSIM↑ | LPIPS↓ | PSNR↑ | SSIM↑ | LPIPS↓ | PSNR↑ | SSIM↑ | LPIPS↓ |
| 032dee9fb0 | 17.43 | 0.510 | 0.447 | 17.04 | 0.521 | 0.375 | 17.03 | 0.573 | 0.412 | 23.07 | 0.709 | 0.196 |
| 0569e83fdc | 12.65 | 0.218 | 0.567 | 12.42 | 0.214 | 0.536 | 12.79 | 0.250 | 0.562 | 15.32 | 0.330 | 0.458 |
| 06da796666 | 13.09 | 0.407 | 0.525 | 13.13 | 0.429 | 0.500 | 14.76 | 0.526 | 0.507 | 14.47 | 0.513 | 0.499 |
| 073f5a9b98 | 11.50 | 0.129 | 0.613 | 11.34 | 0.121 | 0.604 | 12.19 | 0.17 | 0.626 | 12.53 | 0.181 | 0.620 |
| 07d9f9724c | 20.37 | 0.680 | 0.321 | 20.06 | 0.706 | 0.269 | 18.95 | 0.662 | 0.303 | 25.59 | 0.860 | 0.123 |
| 0853979305 | 17.24 | 0.530 | 0.442 | 17.33 | 0.549 | 0.391 | 17.55 | 0.578 | 0.419 | 19.51 | 0.625 | 0.355 |
| 093ef327b4 | 13.08 | 0.445 | 0.587 | 12.85 | 0.422 | 0.540 | 13.49 | 0.452 | 0.552 | 14.96 | 0.493 | 0.472 |
| 0a1b7c20a9 | 15.59 | 0.507 | 0.454 | 15.27 | 0.517 | 0.403 | 15.41 | 0.529 | 0.441 | 19.11 | 0.670 | 0.273 |
| 0a485338bb | 15.13 | 0.467 | 0.466 | 15.24 | 0.491 | 0.402 | 15.89 | 0.491 | 0.440 | 20.23 | 0.623 | 0.290 |
| 0bfdd020cf | 11.08 | 0.397 | 0.660 | 11.23 | 0.355 | 0.560 | 14.27 | 0.432 | 0.579 | 14.87 | 0.402 | 0.486 |
| 119fd56d37 | 12.41 | 0.331 | 0.599 | 12.45 | 0.360 | 0.563 | 13.13 | 0.452 | 0.563 | 13.93 | 0.468 | 0.552 |
| 1264931635 | 12.38 | 0.205 | 0.568 | 12.32 | 0.195 | 0.538 | 13.43 | 0.245 | 0.565 | 14.86 | 0.267 | 0.536 |
| 14eb48a50e | 12.13 | 0.396 | 0.554 | 12.12 | 0.428 | 0.498 | 12.77 | 0.491 | 0.521 | 15.65 | 0.558 | 0.437 |
| 15ff83e253 | 13.37 | 0.360 | 0.576 | 13.59 | 0.330 | 0.491 | 14.00 | 0.389 | 0.552 | 15.65 | 0.383 | 0.474 |
| 165f5af8bf | 14.59 | 0.374 | 0.492 | 14.87 | 0.396 | 0.420 | 15.47 | 0.429 | 0.462 | 16.86 | 0.477 | 0.384 |
| 183dd248f6 | 16.49 | 0.522 | 0.498 | 16.36 | 0.527 | 0.453 | 16.29 | 0.559 | 0.472 | 16.80 | 0.571 | 0.458 |
| 1ba74c2267 | 17.91 | 0.606 | 0.491 | 18.43 | 0.636 | 0.403 | 17.97 | 0.666 | 0.431 | 20.34 | 0.728 | 0.345 |
| 1d6a9ed47c | 12.21 | 0.238 | 0.512 | 12.10 | 0.238 | 0.473 | 12.74 | 0.243 | 0.533 | 15.09 | 0.342 | 0.459 |
| 1da888bded | 13.27 | 0.385 | 0.558 | 13.30 | 0.396 | 0.501 | 13.62 | 0.434 | 0.522 | 16.08 | 0.506 | 0.417 |
| 1de58be515 | 19.19 | 0.707 | 0.279 | 20.09 | 0.732 | 0.197 | 18.32 | 0.629 | 0.319 | 24.18 | 0.820 | 0.165 |
| 2385549d39 | 14.95 | 0.388 | 0.457 | 15.30 | 0.406 | 0.364 | 15.13 | 0.378 | 0.438 | 18.42 | 0.529 | 0.292 |
| 26fd23358f | 14.40 | 0.353 | 0.510 | 14.19 | 0.371 | 0.448 | 14.33 | 0.385 | 0.495 | 17.60 | 0.484 | 0.363 |
| 286239bd0d | 11.63 | 0.249 | 0.571 | 11.57 | 0.251 | 0.552 | 12.5 | 0.304 | 0.561 | 13.15 | 0.311 | 0.546 |
| 2991a75d1f | 12.54 | 0.345 | 0.609 | 12.22 | 0.313 | 0.572 | 12.52 | 0.352 | 0.615 | 13.15 | 0.355 | 0.561 |
| 2b65ba886e | 10.72 | 0.232 | 0.570 | 10.57 | 0.231 | 0.557 | 11.67 | 0.272 | 0.562 | 12.27 | 0.313 | 0.540 |
| 2beaca3189 | 12.30 | 0.268 | 0.580 | 11.99 | 0.271 | 0.541 | 12.74 | 0.305 | 0.570 | 14.58 | 0.373 | 0.465 |
| 2cbfe28643 | 16.33 | 0.559 | 0.425 | 16.98 | 0.606 | 0.314 | 16.49 | 0.595 | 0.395 | 20.11 | 0.725 | 0.231 |
| 2f3e1c0f68 | 13.57 | 0.380 | 0.519 | 13.67 | 0.409 | 0.462 | 14.28 | 0.475 | 0.482 | 15.83 | 0.511 | 0.421 |
| 32c2b92fac | 16.87 | 0.610 | 0.482 | 16.75 | 0.637 | 0.401 | 16.48 | 0.662 | 0.455 | 20.56 | 0.734 | 0.324 |
| 341b4ff3df | 15.35 | 0.408 | 0.478 | 15.14 | 0.414 | 0.417 | 15.18 | 0.410 | 0.457 | 18.05 | 0.558 | 0.327 |
| 35317e6219 | 12.02 | 0.269 | 0.586 | 12.10 | 0.283 | 0.549 | 13.86 | 0.381 | 0.529 | 13.98 | 0.388 | 0.518 |
| 35872363e1 | 14.94 | 0.416 | 0.478 | 14.86 | 0.428 | 0.411 | 15.29 | 0.447 | 0.430 | 17.79 | 0.534 | 0.321 |
| 374ffd0c5f | 18.72 | 0.569 | 0.418 | 19.11 | 0.604 | 0.337 | 17.99 | 0.614 | 0.394 | 21.89 | 0.723 | 0.247 |
| 387eeb925b | 13.14 | 0.317 | 0.551 | 12.96 | 0.332 | 0.469 | 13.39 | 0.375 | 0.528 | 15.93 | 0.461 | 0.392 |
| 389a460ca1 | 13.66 | 0.523 | 0.502 | 13.58 | 0.545 | 0.435 | 12.75 | 0.556 | 0.503 | 15.08 | 0.619 | 0.432 |
| 3b16a10ec9 | 12.67 | 0.337 | 0.516 | 12.83 | 0.357 | 0.472 | 13.92 | 0.437 | 0.487 | 14.99 | 0.460 | 0.457 |
| 3b7529dccc | 12.77 | 0.316 | 0.526 | 12.57 | 0.317 | 0.505 | 13.76 | 0.367 | 0.512 | 14.12 | 0.393 | 0.480 |
| 3bb3bb4d3e | 14.81 | 0.561 | 0.458 | 14.92 | 0.594 | 0.393 | 16.06 | 0.658 | 0.375 | 18.12 | 0.710 | 0.297 |
| 3bb894d193 | 15.94 | 0.429 | 0.426 | 15.92 | 0.449 | 0.330 | 16.57 | 0.463 | 0.380 | 17.93 | 0.573 | 0.255 |
| 41036716da | 14.98 | 0.473 | 0.502 | 15.13 | 0.508 | 0.441 | 15.00 | 0.525 | 0.466 | 16.99 | 0.593 | 0.376 |
| 444da1b4a3 | 12.16 | 0.347 | 0.515 | 11.84 | 0.365 | 0.468 | 11.85 | 0.377 | 0.486 | 16.33 | 0.564 | 0.310 |
| 457e9a1ae7 | 14.08 | 0.434 | 0.520 | 13.90 | 0.455 | 0.474 | 14.28 | 0.495 | 0.508 | 16.21 | 0.579 | 0.425 |
| 484c0aca40 | 13.87 | 0.504 | 0.492 | 13.71 | 0.536 | 0.429 | 13.86 | 0.565 | 0.472 | 16.33 | 0.663 | 0.348 |
| 493816813d | 12.17 | 0.232 | 0.563 | 12.18 | 0.207 | 0.529 | 13.65 | 0.262 | 0.537 | 14.82 | 0.282 | 0.526 |
| 4ae797d07b | 13.07 | 0.378 | 0.520 | 12.80 | 0.361 | 0.491 | 13.24 | 0.386 | 0.504 | 15.26 | 0.435 | 0.449 |
| 4ff8650b5c | 14.93 | 0.470 | 0.499 | 14.84 | 0.488 | 0.457 | 14.87 | 0.518 | 0.476 | 16.16 | 0.559 | 0.404 |
| 50c46cf8b8 | 15.72 | 0.520 | 0.502 | 15.74 | 0.554 | 0.459 | 15.75 | 0.604 | 0.478 | 17.55 | 0.640 | 0.406 |
| 513e4ea2e8 | 14.46 | 0.417 | 0.502 | 14.62 | 0.438 | 0.424 | 14.38 | 0.455 | 0.487 | 20.25 | 0.644 | 0.227 |
| 54bf355ca7 | 13.61 | 0.380 | 0.524 | 13.51 | 0.384 | 0.496 | 14.33 | 0.441 | 0.505 | 15.20 | 0.459 | 0.469 |
| 56452d9cd9 | 14.39 | 0.328 | 0.517 | 14.58 | 0.345 | 0.470 | 15.28 | 0.396 | 0.501 | 16.41 | 0.421 | 0.453 |
| 565553aa89 | 13.51 | 0.318 | 0.527 | 13.65 | 0.338 | 0.474 | 14.67 | 0.384 | 0.487 | 16.35 | 0.453 | 0.408 |
| 599ca3e04c | 14.48 | 0.453 | 0.525 | 14.22 | 0.473 | 0.477 | 13.87 | 0.478 | 0.485 | 15.53 | 0.532 | 0.448 |
| 5a27c00f52 | 13.64 | 0.384 | 0.470 | 13.40 | 0.391 | 0.400 | 13.75 | 0.395 | 0.437 | 17.23 | 0.547 | 0.286 |
| 5a69d1027b | 10.68 | 0.192 | 0.661 | 10.83 | 0.168 | 0.604 | 13.23 | 0.262 | 0.629 | 13.42 | 0.240 | 0.568 |
| 5c3af58102 | 17.45 | 0.563 | 0.389 | 17.94 | 0.618 | 0.280 | 17.04 | 0.592 | 0.338 | 22.53 | 0.767 | 0.180 |
| 5f0041e53d | 18.02 | 0.612 | 0.347 | 18.24 | 0.627 | 0.274 | 17.51 | 0.592 | 0.356 | 21.28 | 0.732 | 0.214 |
| 63798f5c6f | 14.65 | 0.518 | 0.500 | 14.75 | 0.563 | 0.418 | 15.02 | 0.587 | 0.457 | 18.39 | 0.696 | 0.337 |
| 669c36225b | 12.39 | 0.370 | 0.530 | 12.20 | 0.390 | 0.490 | 12.70 | 0.445 | 0.500 | 15.46 | 0.543 | 0.384 |
| 66fd66cbed | 13.83 | 0.536 | 0.492 | 13.90 | 0.570 | 0.423 | 13.90 | 0.602 | 0.446 | 15.85 | 0.654 | 0.378 |
| 6d22162561 | 12.44 | 0.352 | 0.573 | 12.52 | 0.365 | 0.529 | 13.47 | 0.445 | 0.543 | 14.42 | 0.474 | 0.468 |
| 6d81c5ab0d | 15.33 | 0.496 | 0.459 | 15.48 | 0.522 | 0.385 | 15.29 | 0.524 | 0.430 | 18.51 | 0.613 | 0.322 |
| 6e11e7f4fe | 17.59 | 0.714 | 0.472 | 17.93 | 0.731 | 0.404 | 18.92 | 0.779 | 0.415 | 18.49 | 0.760 | 0.377 |
| 70eac6ff18 | 13.67 | 0.398 | 0.480 | 13.38 | 0.403 | 0.423 | 13.22 | 0.387 | 0.479 | 16.16 | 0.529 | 0.348 |
| 71b2dc8a2a | 17.08 | 0.613 | 0.454 | 17.34 | 0.651 | 0.367 | 17.50 | 0.673 | 0.373 | 20.64 | 0.762 | 0.238 |
| 75fbbe4673 | 13.76 | 0.362 | 0.524 | 13.48 | 0.369 | 0.469 | 13.94 | 0.398 | 0.504 | 16.10 | 0.466 | 0.406 |
| 7705a2edd0 | 14.24 | 0.400 | 0.559 | 14.90 | 0.435 | 0.482 | 16.32 | 0.564 | 0.513 | 17.47 | 0.576 | 0.467 |
| 7a9f97660b | 15.45 | 0.473 | 0.512 | 15.48 | 0.520 | 0.425 | 15.12 | 0.549 | 0.474 | 18.36 | 0.612 | 0.348 |
| 7da3db9905 | 12.92 | 0.450 | 0.521 | 12.83 | 0.455 | 0.477 | 13.48 | 0.472 | 0.494 | 13.78 | 0.504 | 0.467 |
| 800cf88687 | 13.75 | 0.318 | 0.524 | 13.42 | 0.321 | 0.490 | 13.67 | 0.350 | 0.518 | 15.43 | 0.376 | 0.467 |
| 8324b3ca22 | 12.22 | 0.304 | 0.592 | 12.35 | 0.327 | 0.548 | 13.04 | 0.417 | 0.551 | 13.76 | 0.410 | 0.522 |
| 85cd0e9211 | 14.24 | 0.365 | 0.528 | 14.40 | 0.386 | 0.469 | 14.28 | 0.432 | 0.498 | 16.48 | 0.472 | 0.429 |
| 8b9fb9d9f1 | 13.16 | 0.288 | 0.551 | 13.26 | 0.299 | 0.520 | 13.63 | 0.342 | 0.576 | 15.26 | 0.381 | 0.499 |
| 8cb2e97d26 | 14.88 | 0.382 | 0.521 | 14.90 | 0.404 | 0.424 | 14.38 | 0.427 | 0.494 | 20.39 | 0.617 | 0.300 |
| 8fdc5130f0 | 15.44 | 0.496 | 0.448 | 15.33 | 0.511 | 0.387 | 15.13 | 0.532 | 0.420 | 17.94 | 0.616 | 0.318 |
| 90cb7ef953 | 15.69 | 0.500 | 0.493 | 15.44 | 0.519 | 0.448 | 15.52 | 0.551 | 0.461 | 15.70 | 0.547 | 0.463 |
| 917e9c8985 | 14.17 | 0.387 | 0.477 | 13.90 | 0.395 | 0.436 | 14.27 | 0.418 | 0.469 | 16.48 | 0.529 | 0.345 |

Table 12: Per-scene 3DGS artifact restoration quantitative comparison on DL3DV-Res (Part 1).

| | Artifact | | | Difix3D+ | | | GenFusion | | | GSFixer (Ours) | | |
|---|---|---|---|---|---|---|---|---|---|---|---|---|
| | PSNR↑ | SSIM↑ | LPIPS↓ | PSNR↑ | SSIM↑ | LPIPS↓ | PSNR↑ | SSIM↑ | LPIPS↓ | PSNR↑ | SSIM↑ | LPIPS↓ |
| 91afb9910b | 13.96 | 0.409 | 0.515 | 13.86 | 0.415 | 0.480 | 15.08 | 0.483 | 0.490 | 16.52 | 0.504 | 0.437 |
| 946f49be73 | 15.67 | 0.454 | 0.436 | 16.25 | 0.473 | 0.337 | 16.06 | 0.484 | 0.385 | 18.61 | 0.574 | 0.287 |
| 9641a1ed79 | 15.73 | 0.523 | 0.506 | 16.07 | 0.561 | 0.432 | 15.29 | 0.562 | 0.489 | 17.48 | 0.626 | 0.391 |
| 9c8c0e0fad | 10.43 | 0.246 | 0.697 | 10.36 | 0.200 | 0.635 | 11.78 | 0.267 | 0.669 | 13.96 | 0.262 | 0.569 |
| 9cbc554864 | 12.32 | 0.256 | 0.550 | 12.11 | 0.259 | 0.529 | 12.76 | 0.296 | 0.543 | 14.95 | 0.382 | 0.456 |
| 9e9a89ae6f | 15.54 | 0.627 | 0.412 | 15.38 | 0.640 | 0.340 | 14.37 | 0.638 | 0.412 | 17.25 | 0.703 | 0.319 |
| 9fb0588ff0 | 14.06 | 0.390 | 0.435 | 14.09 | 0.407 | 0.343 | 14.69 | 0.401 | 0.389 | 18.59 | 0.613 | 0.265 |
| a17a984ca9 | 12.76 | 0.338 | 0.524 | 12.63 | 0.351 | 0.489 | 12.90 | 0.375 | 0.517 | 13.24 | 0.395 | 0.473 |
| a401469cb0 | 12.77 | 0.358 | 0.572 | 12.67 | 0.371 | 0.539 | 13.49 | 0.432 | 0.564 | 14.64 | 0.457 | 0.510 |
| a62c330f54 | 11.94 | 0.317 | 0.572 | 11.81 | 0.332 | 0.550 | 13.27 | 0.414 | 0.555 | 13.86 | 0.428 | 0.528 |
| a62f9a1c63 | 13.93 | 0.400 | 0.502 | 14.00 | 0.412 | 0.420 | 13.63 | 0.406 | 0.494 | 17.54 | 0.545 | 0.343 |
| a726c1112a | 12.30 | 0.463 | 0.530 | 13.11 | 0.510 | 0.481 | 14.38 | 0.606 | 0.481 | 14.36 | 0.592 | 0.490 |
| adb95f29c1 | 14.73 | 0.394 | 0.508 | 15.19 | 0.398 | 0.470 | 15.82 | 0.481 | 0.508 | 17.92 | 0.504 | 0.444 |
| adf35184a1 | 15.87 | 0.526 | 0.523 | 15.86 | 0.559 | 0.438 | 15.68 | 0.588 | 0.484 | 20.09 | 0.691 | 0.339 |
| af0d7039e6 | 12.33 | 0.310 | 0.509 | 12.01 | 0.303 | 0.479 | 12.80 | 0.338 | 0.494 | 13.27 | 0.365 | 0.479 |
| b2076bc723 | 11.04 | 0.267 | 0.593 | 10.76 | 0.274 | 0.585 | 11.83 | 0.332 | 0.582 | 12.61 | 0.400 | 0.529 |
| b3bf9079b4 | 16.08 | 0.461 | 0.441 | 16.23 | 0.496 | 0.371 | 15.98 | 0.502 | 0.414 | 18.68 | 0.579 | 0.326 |
| b4f53094fd | 13.97 | 0.384 | 0.487 | 13.75 | 0.388 | 0.429 | 13.60 | 0.380 | 0.487 | 16.73 | 0.506 | 0.349 |
| b5faa2a8ce | 13.90 | 0.429 | 0.537 | 14.29 | 0.443 | 0.469 | 17.50 | 0.566 | 0.468 | 16.88 | 0.560 | 0.434 |
| b6d1134cb0 | 11.97 | 0.256 | 0.559 | 11.74 | 0.262 | 0.524 | 12.31 | 0.274 | 0.562 | 12.71 | 0.303 | 0.511 |
| b92b499c9b | 18.20 | 0.705 | 0.348 | 19.01 | 0.744 | 0.243 | 20.26 | 0.787 | 0.255 | 22.30 | 0.842 | 0.171 |
| ba55c875d2 | 17.50 | 0.612 | 0.394 | 17.57 | 0.633 | 0.312 | 17.87 | 0.653 | 0.333 | 20.59 | 0.721 | 0.248 |
| c076929db6 | 17.44 | 0.608 | 0.373 | 18.14 | 0.630 | 0.291 | 18.54 | 0.640 | 0.295 | 23.28 | 0.738 | 0.196 |
| c37109a55e | 12.10 | 0.247 | 0.547 | 11.94 | 0.253 | 0.516 | 12.83 | 0.280 | 0.545 | 13.39 | 0.298 | 0.508 |
| c37726ce77 | 13.50 | 0.287 | 0.537 | 13.30 | 0.289 | 0.479 | 14.13 | 0.342 | 0.508 | 14.81 | 0.377 | 0.453 |
| c455899acf | 16.43 | 0.539 | 0.465 | 16.11 | 0.533 | 0.417 | 16.31 | 0.569 | 0.438 | 15.94 | 0.555 | 0.437 |
| cbd44beb04 | 14.85 | 0.433 | 0.498 | 14.64 | 0.450 | 0.442 | 14.32 | 0.455 | 0.511 | 18.03 | 0.588 | 0.341 |
| cc08c0bdc3 | 17.30 | 0.530 | 0.413 | 17.85 | 0.555 | 0.324 | 17.08 | 0.545 | 0.411 | 20.60 | 0.655 | 0.290 |
| cd9c981eeb | 12.40 | 0.300 | 0.539 | 12.41 | 0.313 | 0.492 | 12.82 | 0.351 | 0.510 | 14.82 | 0.438 | 0.420 |
| ceb252f5d4 | 12.86 | 0.401 | 0.529 | 12.66 | 0.419 | 0.496 | 13.04 | 0.467 | 0.528 | 14.87 | 0.521 | 0.445 |
| d1b3a0b37a | 13.59 | 0.374 | 0.538 | 13.36 | 0.367 | 0.508 | 14.65 | 0.404 | 0.523 | 13.70 | 0.382 | 0.531 |
| d3812aad53 | 10.99 | 0.341 | 0.576 | 11.03 | 0.362 | 0.540 | 12.07 | 0.440 | 0.554 | 11.90 | 0.408 | 0.555 |
| d3af8212ae | 15.82 | 0.425 | 0.481 | 16.10 | 0.456 | 0.405 | 16.51 | 0.491 | 0.447 | 21.05 | 0.623 | 0.272 |
| d4fbeba016 | 12.10 | 0.171 | 0.565 | 11.74 | 0.164 | 0.540 | 12.45 | 0.199 | 0.560 | 12.86 | 0.206 | 0.556 |
| d8de66037b | 13.47 | 0.320 | 0.520 | 13.12 | 0.321 | 0.492 | 13.70 | 0.376 | 0.503 | 14.60 | 0.395 | 0.456 |
| d904ae2998 | 11.63 | 0.322 | 0.568 | 11.52 | 0.320 | 0.552 | 12.94 | 0.411 | 0.556 | 13.22 | 0.399 | 0.558 |
| d9b6376623 | 12.74 | 0.315 | 0.570 | 12.89 | 0.320 | 0.535 | 13.57 | 0.380 | 0.559 | 14.71 | 0.408 | 0.503 |
| d9f4c746e6 | 15.33 | 0.516 | 0.409 | 15.59 | 0.543 | 0.329 | 16.54 | 0.564 | 0.346 | 19.76 | 0.698 | 0.210 |
| dac9796dd6 | 14.38 | 0.503 | 0.510 | 14.11 | 0.535 | 0.450 | 13.56 | 0.535 | 0.472 | 18.62 | 0.700 | 0.273 |
| dafa9c7cbd | 11.35 | 0.137 | 0.599 | 11.21 | 0.132 | 0.579 | 12.26 | 0.164 | 0.587 | 12.58 | 0.190 | 0.581 |
| ddfdcfdf02 | 12.63 | 0.293 | 0.557 | 12.28 | 0.297 | 0.514 | 12.66 | 0.310 | 0.553 | 14.55 | 0.401 | 0.450 |
| ded5e4b46a | 17.44 | 0.563 | 0.506 | 17.23 | 0.580 | 0.458 | 17.80 | 0.628 | 0.479 | 18.78 | 0.658 | 0.425 |
| df04f58064 | 14.02 | 0.277 | 0.529 | 13.91 | 0.285 | 0.481 | 13.90 | 0.286 | 0.532 | 17.45 | 0.442 | 0.376 |
| df29c22586 | 15.52 | 0.402 | 0.425 | 15.60 | 0.421 | 0.338 | 15.80 | 0.427 | 0.397 | 20.31 | 0.589 | 0.281 |
| df4f9d9a0a | 13.12 | 0.473 | 0.321 | 13.26 | 0.504 | 0.430 | 14.42 | 0.563 | 0.452 | 17.58 | 0.662 | 0.340 |
| e5684b3292 | 13.90 | 0.415 | 0.545 | 13.93 | 0.447 | 0.497 | 13.25 | 0.506 | 0.537 | 15.40 | 0.550 | 0.453 |
| e78f8cebd2 | 12.51 | 0.206 | 0.562 | 12.45 | 0.209 | 0.534 | 13.32 | 0.241 | 0.546 | 14.71 | 0.296 | 0.491 |
| e8ce51b6ab | 14.38 | 0.491 | 0.499 | 14.17 | 0.513 | 0.438 | 14.27 | 0.529 | 0.486 | 18.52 | 0.687 | 0.284 |
| e9360e7a89 | 12.13 | 0.319 | 0.554 | 11.84 | 0.319 | 0.521 | 12.31 | 0.335 | 0.565 | 12.67 | 0.362 | 0.516 |
| eb4cf52988 | 13.75 | 0.417 | 0.543 | 13.53 | 0.431 | 0.518 | 14.12 | 0.476 | 0.532 | 15.67 | 0.526 | 0.478 |
| ec1e44d4dc | 13.81 | 0.452 | 0.534 | 13.50 | 0.481 | 0.489 | 13.51 | 0.506 | 0.497 | 16.37 | 0.606 | 0.353 |
| ec305787b7 | 13.68 | 0.476 | 0.484 | 13.77 | 0.509 | 0.415 | 14.16 | 0.569 | 0.389 | 14.99 | 0.575 | 0.375 |
| ed16328235 | 14.50 | 0.214 | 0.549 | 15.04 | 0.217 | 0.500 | 15.89 | 0.277 | 0.551 | 17.07 | 0.293 | 0.483 |
| ef59aac437 | 17.15 | 0.331 | 0.515 | 17.32 | 0.338 | 0.443 | 17.71 | 0.381 | 0.498 | 19.97 | 0.465 | 0.361 |
| f004c810d9 | 14.48 | 0.594 | 0.405 | 14.51 | 0.624 | 0.317 | 14.21 | 0.631 | 0.343 | 19.79 | 0.760 | 0.210 |
| f477ffc4b3 | 14.75 | 0.528 | 0.490 | 14.38 | 0.551 | 0.421 | 13.7 | 0.534 | 0.461 | 17.21 | 0.635 | 0.336 |
| f71ac346cd | 13.23 | 0.230 | 0.534 | 13.06 | 0.224 | 0.499 | 13.72 | 0.254 | 0.529 | 15.63 | 0.328 | 0.440 |
| f7aaea9ac6 | 13.11 | 0.247 | 0.529 | 13.14 | 0.253 | 0.473 | 13.76 | 0.260 | 0.530 | 16.34 | 0.388 | 0.416 |
| fb2c0499c2 | 14.93 | 0.442 | 0.469 | 15.61 | 0.481 | 0.375 | 16.38 | 0.530 | 0.381 | 19.14 | 0.608 | 0.284 |
| fb3b73f1d3 | 11.78 | 0.132 | 0.622 | 11.72 | 0.125 | 0.610 | 12.80 | 0.180 | 0.630 | 13.41 | 0.189 | 0.588 |
| ff59239865 | 12.35 | 0.432 | 0.533 | 12.81 | 0.420 | 0.481 | 14.28 | 0.455 | 0.493 | 17.72 | 0.516 | 0.395 |
| average | 14.12 | 0.405 | 0.509 | 14.14 | 0.419 | 0.455 | 14.56 | 0.453 | 0.486 | 16.72 | 0.520 | 0.399 |

Table 13: Per-scene 3DGS artifact restoration quantitative comparison on DL3DV-Res (Part 2).

| | Difix3D+ | | | GenFusion | | | GSFixer (Ours) | | |
|---|---|---|---|---|---|---|---|---|---|
| | PSNR↑ | SSIM↑ | LPIPS↓ | PSNR↑ | SSIM↑ | LPIPS↓ | PSNR↑ | SSIM↑ | LPIPS↓ |
| 0bfdd020cf | 15.96 | 0.498 | 0.494 | 15.16 | 0.511 | 0.510 | 17.04 | 0.526 | 0.476 |
| 2beaca3189 | 14.33 | 0.401 | 0.423 | 13.31 | 0.380 | 0.546 | 14.78 | 0.457 | 0.526 |
| 5a69d1027b | 14.70 | 0.359 | 0.452 | 15.49 | 0.407 | 0.514 | 16.56 | 0.435 | 0.465 |
| 9c8c0e0fad | 11.50 | 0.260 | 0.605 | 12.10 | 0.306 | 0.648 | 13.77 | 0.343 | 0.623 |
| 032dee9fb0 | 15.66 | 0.528 | 0.509 | 12.74 | 0.506 | 0.489 | 16.57 | 0.565 | 0.455 |
| 85cd0e9211 | 17.78 | 0.550 | 0.383 | 16.96 | 0.580 | 0.437 | 18.03 | 0.578 | 0.441 |
| 91afb9910b | 14.09 | 0.508 | 0.518 | 15.14 | 0.576 | 0.481 | 16.20 | 0.589 | 0.485 |
| 165f5af8bf | 16.49 | 0.476 | 0.395 | 15.63 | 0.496 | 0.458 | 16.85 | 0.516 | 0.439 |
| 374ffd0c5f | 18.62 | 0.639 | 0.418 | 18.72 | 0.669 | 0.404 | 20.03 | 0.692 | 0.393 |
| 457e9a1ae7 | 16.28 | 0.583 | 0.458 | 17.13 | 0.640 | 0.417 | 18.33 | 0.660 | 0.416 |
| 669c36225b | 12.01 | 0.449 | 0.576 | 11.78 | 0.496 | 0.539 | 13.28 | 0.503 | 0.527 |
| 9641a1ed79 | 15.37 | 0.610 | 0.621 | 15.19 | 0.615 | 0.517 | 16.46 | 0.612 | 0.495 |
| 56452d9cd9 | 13.76 | 0.400 | 0.549 | 13.74 | 0.418 | 0.526 | 14.93 | 0.420 | 0.513 |
| 493816813d | 14.91 | 0.358 | 0.404 | 14.91 | 0.414 | 0.485 | 16.49 | 0.612 | 0.496 |
| 0853979305 | 18.31 | 0.630 | 0.368 | 17.21 | 0.653 | 0.417 | 18.62 | 0.671 | 0.394 |
| adb95f29c1 | 14.07 | 0.427 | 0.538 | 13.09 | 0.452 | 0.565 | 15.24 | 0.485 | 0.550 |
| b3bf9079b4 | 15.43 | 0.555 | 0.374 | 15.07 | 0.583 | 0.394 | 16.79 | 0.590 | 0.396 |
| ba55c875d2 | 17.18 | 0.613 | 0.386 | 16.02 | 0.600 | 0.421 | 18.10 | 0.635 | 0.397 |
| d1b3a0b37a | 13.90 | 0.443 | 0.527 | 13.39 | 0.456 | 0.523 | 14.78 | 0.486 | 0.520 |
| d904ae2998 | 12.94 | 0.439 | 0.647 | 10.92 | 0.344 | 0.593 | 13.06 | 0.416 | 0.568 |
| dac9796dd6 | 13.90 | 0.561 | 0.518 | 14.29 | 0.559 | 0.453 | 15.79 | 0.610 | 0.443 |
| dafa9c7cbd | 11.68 | 0.169 | 0.497 | 11.81 | 0.224 | 0.643 | 12.64 | 0.250 | 0.624 |
| df29c22586 | 19.31 | 0.538 | 0.238 | 19.84 | 0.585 | 0.370 | 21.13 | 0.633 | 0.318 |
| e9360e7a89 | 14.50 | 0.463 | 0.490 | 13.52 | 0.446 | 0.508 | 14.82 | 0.492 | 0.526 |
| ec305787b7 | 14.23 | 0.587 | 0.515 | 13.37 | 0.511 | 0.435 | 15.44 | 0.640 | 0.413 |
| ed16328235 | 14.83 | 0.281 | 0.485 | 13.58 | 0.306 | 0.606 | 15.83 | 0.355 | 0.573 |
| f004c810d9 | 17.53 | 0.729 | 0.314 | 17.76 | 0.757 | 0.321 | 18.85 | 0.763 | 0.325 |
| ff59239865 | 12.56 | 0.416 | 0.547 | 12.09 | 0.443 | 0.583 | 13.51 | 0.467 | 0.574 |
| average | 15.07 | 0.4816 | 0.473 | 14.64 | 0.498 | 0.493 | 16.21 | 0.536 | 0.478 |

Table 14: Comparison of per-scene 3D reconstruction results of DL3DV with 3 input views.

| | Difix3D+ | | | GenFusion | | | GSFixer (Ours) | | |
|---|---|---|---|---|---|---|---|---|---|
| | PSNR↑ | SSIM↑ | LPIPS↓ | PSNR↑ | SSIM↑ | LPIPS↓ | PSNR↑ | SSIM↑ | LPIPS↓ |
| 0bfdd020cf | 17.54 | 0.547 | 0.394 | 17.39 | 0.584 | 0.405 | 18.21 | 0.582 | 0.399 |
| 2beaca3189 | 17.56 | 0.592 | 0.250 | 18.39 | 0.638 | 0.359 | 18.37 | 0.627 | 0.361 |
| 5a69d1027b | 17.35 | 0.471 | 0.322 | 16.97 | 0.501 | 0.407 | 18.47 | 0.516 | 0.365 |
| 9c8c0e0fad | 15.15 | 0.342 | 0.425 | 14.69 | 0.378 | 0.535 | 16.14 | 0.399 | 0.472 |
| 032dee9fb0 | 20.48 | 0.690 | 0.265 | 19.65 | 0.727 | 0.293 | 21.13 | 0.722 | 0.285 |
| 85cd0e9211 | 20.60 | 0.673 | 0.258 | 20.14 | 0.692 | 0.337 | 22.32 | 0.754 | 0.266 |
| 91afb9910b | 17.56 | 0.622 | 0.346 | 18.53 | 0.673 | 0.366 | 18.64 | 0.672 | 0.365 |
| 165f5af8bf | 18.06 | 0.551 | 0.302 | 17.76 | 0.570 | 0.374 | 18.71 | 0.576 | 0.349 |
| 374ffd0c5f | 23.98 | 0.807 | 0.195 | 24.29 | 0.827 | 0.224 | 24.50 | 0.819 | 0.227 |
| 457e9a1ae7 | 19.48 | 0.694 | 0.347 | 20.48 | 0.752 | 0.322 | 20.93 | 0.735 | 0.343 |
| 669c36225b | 15.42 | 0.566 | 0.400 | 16.24 | 0.300 | 0.396 | 16.64 | 0.612 | 0.412 |
| 9641a1ed79 | 18.05 | 0.658 | 0.470 | 18.17 | 0.668 | 0.418 | 18.47 | 0.651 | 0.419 |
| 56452d9cd9 | 16.35 | 0.422 | 0.409 | 17.70 | 0.511 | 0.420 | 18.84 | 0.562 | 0.373 |
| 493816813d | 16.89 | 0.482 | 0.326 | 16.93 | 0.515 | 0.406 | 17.54 | 0.522 | 0.401 |
| 0853979305 | 22.35 | 0.771 | 0.212 | 22.39 | 0.807 | 0.245 | 22.71 | 0.789 | 0.260 |
| adb95f29c1 | 16.78 | 0.516 | 0.435 | 15.73 | 0.526 | 0.482 | 18.87 | 0.583 | 0.383 |
| b3bf9079b4 | 18.84 | 0.662 | 0.253 | 18.29 | 0.696 | 0.296 | 18.56 | 0.676 | 0.301 |
| ba55c875d2 | 20.78 | 0.742 | 0.227 | 20.18 | 0.764 | 0.273 | 21.37 | 0.751 | 0.265 |
| d1b3a0b37a | 16.22 | 0.516 | 0.435 | 16.82 | 0.566 | 0.426 | 16.66 | 0.538 | 0.450 |
| d904ae2998 | 14.36 | 0.480 | 0.541 | 13.10 | 0.446 | 0.533 | 13.97 | 0.480 | 0.526 |
| dac9796dd6 | 19.83 | 0.735 | 0.285 | 20.63 | 0.767 | 0.278 | 20.87 | 0.768 | 0.279 |
| dafa9c7cbd | 13.32 | 0.226 | 0.418 | 13.29 | 0.270 | 0.573 | 14.21 | 0.279 | 0.541 |
| df29c22586 | 24.24 | 0.713 | 0.150 | 24.64 | 0.737 | 0.256 | 24.57 | 0.724 | 0.216 |
| e9360e7a89 | 15.80 | 0.540 | 0.372 | 15.63 | 0.559 | 0.426 | 16.09 | 0.569 | 0.430 |
| ec305787b7 | 18.53 | 0.701 | 0.310 | 19.27 | 0.733 | 0.299 | 19.01 | 0.723 | 0.309 |
| ed16328235 | 16.18 | 0.358 | 0.382 | 16.29 | 0.398 | 0.504 | 16.50 | 0.400 | 0.502 |
| f004c810d9 | 22.77 | 0.859 | 0.157 | 22.94 | 0.872 | 0.202 | 23.62 | 0.874 | 0.194 |
| ff59239865 | 16.89 | 0.549 | 0.339 | 17.59 | 0.611 | 0.403 | 19.05 | 0.600 | 0.378 |
| average | 18.26 | 0.5895 | 0.329 | 18.36 | 0.610 | 0.374 | 19.11 | 0.625 | 0.360 |

Table 15: Comparison of per-scene 3D reconstruction results of DL3DV with 6 input views.

| | Difix3D+ | | | GenFusion | | | GSFixer (Ours) | | |
|---|---|---|---|---|---|---|---|---|---|
| | PSNR↑ | SSIM↑ | LPIPS↓ | PSNR↑ | SSIM↑ | LPIPS↓ | PSNR↑ | SSIM↑ | LPIPS↓ |
| 0bfdd020cf | 19.15 | 0.582 | 0.320 | 19.70 | 0.635 | 0.366 | 20.03 | 0.605 | 0.352 |
| 2beaca3189 | 19.79 | 0.694 | 0.185 | 20.52 | 0.725 | 0.284 | 20.20 | 0.706 | 0.286 |
| 5a69d1027b | 18.69 | 0.542 | 0.270 | 18.80 | 0.578 | 0.356 | 19.83 | 0.581 | 0.316 |
| 9c8c0e0fad | 16.76 | 0.411 | 0.380 | 16.67 | 0.442 | 0.482 | 17.35 | 0.442 | 0.436 |
| 032dee9fb0 | 22.93 | 0.787 | 0.169 | 22.73 | 0.829 | 0.211 | 22.89 | 0.801 | 0.209 |
| 85cd0e9211 | 22.88 | 0.762 | 0.174 | 22.60 | 0.778 | 0.267 | 22.48 | 0.757 | 0.262 |
| 91afb9910b | 20.45 | 0.715 | 0.248 | 20.28 | 0.737 | 0.301 | 20.51 | 0.733 | 0.295 |
| 165f5af8bf | 19.41 | 0.597 | 0.255 | 19.12 | 0.617 | 0.330 | 19.67 | 0.611 | 0.316 |
| 374ffd0c5f | 26.67 | 0.871 | 0.120 | 27.01 | 0.886 | 0.163 | 26.67 | 0.874 | 0.160 |
| 457e9a1ae7 | 21.66 | 0.761 | 0.262 | 22.05 | 0.800 | 0.281 | 22.21 | 0.777 | 0.299 |
| 669c36225b | 17.03 | 0.636 | 0.324 | 17.91 | 0.690 | 0.336 | 17.60 | 0.656 | 0.362 |
| 9641a1ed79 | 18.85 | 0.675 | 0.419 | 19.30 | 0.702 | 0.389 | 19.32 | 0.676 | 0.389 |
| 56452d9cd9 | 17.32 | 0.473 | 0.393 | 19.01 | 0.594 | 0.370 | 19.29 | 0.561 | 0.372 |
| 493816813d | 18.35 | 0.563 | 0.249 | 18.11 | 0.581 | 0.338 | 18.88 | 0.596 | 0.324 |
| 0853979305 | 24.54 | 0.835 | 0.152 | 24.66 | 0.856 | 0.192 | 24.86 | 0.841 | 0.199 |
| adb95f29c1 | 18.29 | 0.557 | 0.343 | 18.17 | 0.598 | 0.399 | 18.88 | 0.584 | 0.383 |
| b3bf9079b4 | 19.37 | 0.717 | 0.212 | 19.76 | 0.762 | 0.247 | 19.87 | 0.737 | 0.254 |
| ba55c875d2 | 22.98 | 0.808 | 0.170 | 22.79 | 0.825 | 0.207 | 23.79 | 0.818 | 0.200 |
| d1b3a0b37a | 18.41 | 0.607 | 0.315 | 18.97 | 0.660 | 0.332 | 19.06 | 0.623 | 0.355 |
| d904ae2998 | 16.80 | 0.548 | 0.408 | 16.03 | 0.548 | 0.443 | 16.71 | 0.553 | 0.430 |
| dac9796dd6 | 22.06 | 0.807 | 0.207 | 22.00 | 0.815 | 0.232 | 22.42 | 0.812 | 0.231 |
| dafa9c7cbd | 14.33 | 0.276 | 0.385 | 13.92 | 0.297 | 0.533 | 14.80 | 0.309 | 0.512 |
| df29c22586 | 25.66 | 0.759 | 0.115 | 26.14 | 0.782 | 0.228 | 26.01 | 0.774 | 0.172 |
| e9360e7a89 | 17.95 | 0.626 | 0.289 | 18.66 | 0.668 | 0.339 | 18.15 | 0.643 | 0.352 |
| ec305787b7 | 21.07 | 0.776 | 0.203 | 21.65 | 0.806 | 0.220 | 21.54 | 0.790 | 0.224 |
| ed16328235 | 18.44 | 0.460 | 0.295 | 17.98 | 0.477 | 0.417 | 18.18 | 0.467 | 0.414 |
| f004c810d9 | 25.11 | 0.899 | 0.126 | 25.50 | 0.911 | 0.157 | 25.40 | 0.906 | 0.157 |
| ff59239865 | 18.37 | 0.613 | 0.272 | 18.88 | 0.658 | 0.369 | 20.13 | 0.654 | 0.333 |
| average | 20.12 | 0.656 | 0.259 | 20.32 | 0.688 | 0.314 | 20.60 | 0.675 | 0.307 |

Table 16: Comparison of per-scene 3D reconstruction results of DL3DV with 9 input views.

| | Difix3D+ | | | GenFusion | | | GSFixer (Ours) | | |
|---|---|---|---|---|---|---|---|---|---|
| | PSNR↑ | SSIM↑ | LPIPS↓ | PSNR↑ | SSIM↑ | LPIPS↓ | PSNR↑ | SSIM↑ | LPIPS↓ |
| 3 Views | | | | | | | | | |
| bicycle | 13.02 | 0.177 | 0.632 | 14.75 | 0.247 | 0.632 | 15.80 | 0.284 | 0.613 |
| bonsai | 13.86 | 0.397 | 0.549 | 13.87 | 0.408 | 0.549 | 13.42 | 0.434 | 0.554 |
| counter | 13.96 | 0.394 | 0.519 | 15.20 | 0.470 | 0.519 | 15.25 | 0.472 | 0.505 |
| flowers | 11.81 | 0.154 | 0.693 | 12.86 | 0.202 | 0.693 | 13.80 | 0.288 | 0.654 |
| garden | 14.34 | 0.258 | 0.573 | 16.13 | 0.289 | 0.573 | 17.42 | 0.326 | 0.537 |
| kitchen | 15.56 | 0.371 | 0.537 | 16.14 | 0.419 | 0.537 | 16.61 | 0.424 | 0.503 |
| room | 14.15 | 0.465 | 0.427 | 16.23 | 0.568 | 0.427 | 15.54 | 0.540 | 0.441 |
| stump | 15.35 | 0.215 | 0.620 | 16.51 | 0.296 | 0.620 | 17.62 | 0.320 | 0.603 |
| treehill | 13.19 | 0.253 | 0.649 | 13.57 | 0.311 | 0.649 | 15.03 | 0.319 | 0.619 |
| average | 13.92 | 0.298 | 0.578 | 15.03 | 0.357 | 0.578 | 15.61 | 0.370 | 0.559 |
| 6 Views | | | | | | | | | |
| bicycle | 15.58 | 0.253 | 0.594 | 16.16 | 0.292 | 0.570 | 17.32 | 0.302 | 0.549 |
| bonsai | 16.30 | 0.525 | 0.427 | 17.08 | 0.545 | 0.426 | 17.29 | 0.556 | 0.440 |
| counter | 16.09 | 0.490 | 0.413 | 17.06 | 0.545 | 0.426 | 17.12 | 0.530 | 0.425 |
| flowers | 12.66 | 0.178 | 0.572 | 13.69 | 0.224 | 0.632 | 14.10 | 0.216 | 0.594 |
| garden | 17.58 | 0.360 | 0.363 | 18.62 | 0.396 | 0.452 | 18.98 | 0.400 | 0.429 |
| kitchen | 17.95 | 0.534 | 0.310 | 18.62 | 0.556 | 0.388 | 18.78 | 0.555 | 0.371 |
| room | 15.07 | 0.525 | 0.460 | 17.73 | 0.628 | 0.387 | 16.76 | 0.606 | 0.403 |
| stump | 17.51 | 0.284 | 0.485 | 17.77 | 0.321 | 0.570 | 18.65 | 0.328 | 0.543 |
| treehill | 14.70 | 0.293 | 0.588 | 15.36 | 0.353 | 0.585 | 16.46 | 0.341 | 0.546 |
| average | 15.94 | 0.382 | 0.468 | 16.90 | 0.430 | 0.494 | 17.27 | 0.426 | 0.478 |
| 9 Views | | | | | | | | | |
| bicycle | 16.99 | 0.288 | 0.498 | 16.89 | 0.313 | 0.541 | 17.89 | 0.322 | 0.511 |
| bonsai | 18.54 | 0.661 | 0.343 | 19.43 | 0.661 | 0.343 | 19.53 | 0.654 | 0.356 |
| counter | 17.42 | 0.552 | 0.352 | 18.18 | 0.607 | 0.368 | 18.82 | 0.600 | 0.362 |
| flowers | 13.97 | 0.218 | 0.499 | 14.50 | 0.253 | 0.593 | 14.98 | 0.244 | 0.534 |
| garden | 19.18 | 0.453 | 0.282 | 19.79 | 0.471 | 0.393 | 20.10 | 0.473 | 0.367 |
| kitchen | 20.08 | 0.610 | 0.243 | 20.56 | 0.635 | 0.316 | 20.58 | 0.623 | 0.314 |
| room | 17.96 | 0.626 | 0.343 | 19.72 | 0.700 | 0.317 | 18.96 | 0.666 | 0.333 |
| stump | 18.50 | 0.340 | 0.425 | 19.09 | 0.376 | 0.522 | 19.59 | 0.378 | 0.496 |
| treehill | 15.25 | 0.322 | 0.537 | 16.42 | 0.387 | 0.566 | 17.24 | 0.366 | 0.507 |
| average | 17.54 | 0.452 | 0.391 | 18.29 | 0.489 | 0.507 | 18.63 | 0.481 | 0.420 |

Table 17: Per-scene 3D sparse view reconstruction quantitative comparison on Mip-NeRF 360.