# OpenReview forum: "GSFixer: Improving 3D Gaussian Splatting with Reference-Guided Video Diffusion Priors"
_ICLR.cc/2026/Conference — Submitted to ICLR 2026_

### Official Review · Reviewer_YpNj · 2025-10-27

**Soundness:** 3
**Presentation:** 4
**Contribution:** 2
**Rating:** 6
**Confidence:** 5

**Summary:**

This paper introduces VGGT and DINOv2 as additional conditions to enhance the performance of leveraging video diffusion prior to 3D Gaussian Splatting. It also proposes a novel reference-guided trajectory strategy that improves view-angle coverage during sampling, thereby enhancing the final reconstruction quality. Extensive experimental results demonstrate that the proposed method achieves strong and consistent performance improvements.

**Strengths:**

1. This paper introduces features from VGGT and DINOv2 as additional conditioning inputs for video diffusion within the 3D Gaussian Splatting framework. The idea is novel and brings clear performance improvements over existing methods.

2.The proposed Reference-Guided Trajectory provides a valuable insight into how to generate more suitable camera poses for rendering, which can help produce better inputs for the diffusion model and ultimately improve 3D Gaussian Splatting performance. This is an interesting and meaningful direction.

3.The paper is clearly written, and the extensive experiments convincingly demonstrate that the proposed method is effective.

**Weaknesses:**

1. While the method is technically sound, I find the contribution somewhat incremental. The overall framework is similar to previous approaches such as Difix3D+, which also progressively integrate diffusion priors to improve 3D Gaussian Splatting. Using video diffusion models to enhance 3DGS is no longer new, and incorporating 3D information into diffusion models has already been explored in works like ReconX. Thus, the novelty of this paper is somewhat limited.

2. I would have liked to see a deeper discussion on why the diffusion model benefits from conditioning on features from DINOv2 or VGGT since it is the paper's key contribution. For instance, if we consider using geometry foundation models such as VGGT  MapAnything, or MVSanywhere [3]  to generate depth maps as a supervision for 3DGS, would this be more effective than using them as conditioning inputs to the diffusion model? Addressing such questions could provide a stronger understanding of the method’s design choices and its underlying reason.

3. Difix3D+ already mentioned that its diffusion model supports multiple reference views, and the recent paper FlowR [4] also demonstrates that using multi-view reference information can effectively improve 3D Gaussian Splatting. Therefore, the core idea of reference-guided is not entirely novel in current stage.

Below are several relevant citations I included in my review that were not discussed in the paper.

[1] Liu, F., Sun, W., Wang, H., Wang, Y., Sun, H., Ye, J., ... & Duan, Y. (2024). Reconx: Reconstruct any scene from sparse views with video diffusion model. arXiv preprint arXiv:2408.16767.

[2] Keetha, N., Müller, N., Schönberger, J., Porzi, L., Zhang, Y., Fischer, T., ... & Kontschieder, P. (2025). MapAnything: Universal feed-forward metric 3D reconstruction. arXiv preprint arXiv:2509.13414.

[3] Izquierdo, S., Sayed, M., Firman, M., Garcia-Hernando, G., Turmukhambetov, D., Civera, J., ... & Watson, J. (2025). MVSAnywhere: Zero-Shot Multi-View Stereo. In Proceedings of the Computer Vision and Pattern Recognition Conference (pp. 11493-11504).

[4] Fischer, T., Bulò, S. R., Yang, Y. H., Keetha, N., Porzi, L., Müller, N., ... & Kontschieder, P. (2025). Flowr: Flowing from sparse to dense 3d reconstructions. In Proceedings of the IEEE/CVF International Conference on Computer Vision (pp. 27702-27712).

**Questions:**

1. Why is a new benchmark necessary, and what limitations of existing datasets does it address?
2. What are the training settings for your 3D Gaussian Splatting, including whether the points are initialized randomly or from colmap, and  the learning rate for 3DGS, difix3D and your method?
3. For Difix3D+, did you evaluate with the post-rendering refinement step, or report the score without it?

---

> ### Author Response · Authors · 2025-11-25
> **Official Response by Authors -- Part 1**
>
> We would like to sincerely thank the reviewer for the valuable and constructive comments on our work. To effectively respond to all your concerns, we first provide all the additional experiments requested by the reviewer, and then respond to all remaining comments. We take every comment seriously and hope our response can address the reviewer’s concerns.
>
> **Q1: I would have liked to see a deeper discussion on why the diffusion model benefits from conditioning on features from DINOv2 or VGGT since it is the paper's key contribution. For instance, if we consider using geometry foundation models such as VGGT MapAnything, or MVSanywhere to generate depth maps as a supervision for 3DGS, would this be more effective than using them as conditioning inputs to the diffusion model? Addressing such questions could provide a stronger understanding of the method’s design choices and its underlying reason.**
>
> Thanks for your constructive suggestions. Following your recommendation, we conduct additional experiments that training our GSFixer conditioning on the depthmap of VGGT and evaluating the performance on our DL3DV-Res dataset. The results in the below table show that conditioning on explicit depthmap of VGGT results in significantly lower performance compared to our proposed feature-based approach.
>
> Explicit depth maps generated from sparse views using VGGT may contain noise or structural errors. When used as a direct condition, these errors may propagate into the restoration process, forcing the model to adhere to incorrect geometry. In contrast, conditioning on latent features of VGGT allows the model to leverage rich geometric information. This implicit guidance enables the diffusion model to utilize geometric cues without overfitting to the inaccuracies present in depthmap. We also provide visual comparisons in **Figure 13** of the revised manuscript, which further demonstrate that our implicit feature-based conditioning yields superior robustness compared to explicit depthmap conditioning.
>
> | Method       | PSNR | SSIM | LPIPS |
> |--------------|--------|--------|---------|
> | depthmap     | 15.13  | 0.485  | 0.431   |
> | 3D features  | 16.48  | 0.516  | 0.409   |

---

> ### Author Response · Authors · 2025-11-25
> **Official Response by Authors -- Part 2**
>
> **Q2 & Q3: Difix3D+ already mentioned that its diffusion model supports multiple reference views, and the recent paper FlowR also demonstrates that using multi-view reference information can effectively improve 3D Gaussian Splatting. Therefore, the core idea of reference-guided is not entirely novel in current stage. While the method is technically sound, I find the contribution somewhat incremental. The overall framework is similar to previous approaches such as Difix3D+, which also progressively integrate diffusion priors to improve 3D Gaussian Splatting. Using video diffusion models to enhance 3DGS is no longer new, and incorporating 3D information into diffusion models has already been explored in works like ReconX. Thus, the novelty of this paper is somewhat limited.**
>
>
> Thanks for your thoughtful comments and for highlighting these relevant works. We would like to clarify several substantive distinctions between our method and Difix3D+, FlowR, and ReconX. We will include the discussion in the revised related-work section.
> - **Clarification on Difix3D+**: Difix3D+ conditions an image diffusion model on a single reference view, while reference-guided conditioning for video diffusion in this task remains unexplored. Although Difix3D+ briefly mentions the possibility of using multiple reference views, its official implementation and results rely on **using either no reference or one nearest reference view**. It lacks a dedicated mechanism to effectively fuse information from multiple views. To verify this empirically, we extend Difix3D+ to use two nearest reference views on the Mip-NeRF 360 dataset under 3-view setting. As shown in the table below, Difix3D+ with **two nearest reference views** performance slightly **degrades** compared to the single-view baseline. This indicates that Difix3D+ cannot be straightforwardly extended to multi-reference conditioning. In contrast, our **GSFixer is explicitly designed for multi-view reference fusion**, leveraging both **3D geometry (VGGT) and 2D semantic features (DINOv2)** from multiple clean reference views via cross-attention. This **multi-view, multi-modal conditions** injecting remains underexplored in existing literature and **effectively addresses the consistency issues where Difix3D+ struggles**.
>
> | Method               | PSNR ↑ | SSIM ↑ | LPIPS ↓ |
> |----------------------|--------|--------|---------|
> | Difix3D+ (no ref)    | 13.61  | 0.274  | 0.599   |
> | Difix3D+ (one ref)   | 13.92  | 0.298  | 0.578   |
> | Difix3D+ (two ref)   | 13.86  | 0.294  | 0.595   |
>
> - **Distinction from FlowR**: we would like to clarify that the recent paper FlowR condition its flow matching model on **multi-view noise novel views**, which closely follows the GenFusion paradigm of conditioning on artifact frames to fix artifacts. Because FlowR does not release its code, a direct comparison is infeasible. Based on the paper, we can observe that FlowR **does not use clean input views** as conditioning signals. In our GSFixer, **the multi-view reference information** specifically refers to **multi-view clean input views**. Our reference-guided paradigm uses additional clean semantic and geometric features of clean reference views to restore the artifact novel views, which remain unexplored in existing methods.
>
> - **Distinction from ReconX**: ReconX has not open-sourced its implementation, preventing a quantitative comparison. According to the paper, ReconX relies on **the explicit pointmap output of DUSt3R** [1] as 3D condition on U-Net based video diffusion model. As discussed in our response to Q1, explicit geometric outputs (whether depthmap from VGGT or pointmap from DUSt3R) **suffer from noise or structural errors** in sparse-view settings. In contrast, our GSFixer is the first to use **implicit latent features** from foundation models (fusion of VGGT geometric tokens and DINOv2 semantic tokens), enabling our DiT-based video diffusion model to leverage rich geometric and semantic features without being constrained by the inaccuracies inherent in explicit depthmaps or pointmaps.
>
> [1] DUSt3R: Geometric 3d vision made easy. CVPR 2024.

---

> ### Author Response · Authors · 2025-11-25
> **Official Response by Authors -- Part 3**
>
> **Q4: For Difix3D+, did you evaluate with the post-rendering refinement step, or report the score without it?**
>
> We clarify that the results reported for Difix3D+ in our main paper were obtained without the post-rendering refinement step.  For completeness, we provide in the table below the results of Difix3D+ with and without post-rendering refinement on the Mip-NeRF 360 dataset. The refinement step is effective for dense-view settings, which are demonstrated in Difix3D+ paper. However, in the sparse-view settings, the final 3DGS renders contain severe artifacts and large missing regions. Applying the additional post-rendering refinement step tends to amplifies these ambiguous or distorted geometry regions, leading to lower fidelity compared to the variant without the refinement. This phenomenon is consistent with the observations mentioned in the recent paper FixingGS [1], and we also provide qualitative results in the **Figure 15** of our revised manuscript.
>
> [1] FixingGS: Enhancing 3D gaussian splatting via training-free score distillation. ArXiv 2025.
>
> | Method               | PSNR (3-view) | PSNR (6-view) | PSNR (9-view) | SSIM (3-view) | SSIM (6-view) | SSIM (9-view) | LPIPS (3-view) | LPIPS (6-view) | LPIPS (9-view) |
> |----------------------|------------|------------|------------|------------|------------|------------|-------------|-------------|-------------|
> | Difix3D+ w/o post refine   | 13.92      | 15.94      | 17.54      | 0.298      | 0.382      | 0.452      | 0.578       | 0.468       | 0.391       |
> | Difix3D+ w post refine     | 13.77      | 15.70      | 17.20      | 0.263      | 0.342      | 0.401      | 0.549       | 0.460       | 0.411       |
>
>
> ---
>
> **Q5: Why is a new benchmark necessary, and what limitations of existing datasets does it address?**
>
> The introduction of DL3DV-Res is necessary because existing datasets (e.g., Mip-NeRF 360 and DL3DV-Benchmark) are designed for traditional reconstruction tasks. Our proposed method and the recent generative reconstruction approaches rely on a critical intermediate restoration step, which lacks a standardized evaluation protocol.
>
> DL3DV-Res addresses the following specific limitations:
> - **Decoupling Restoration from Reconstruction:**  Existing datasets provide only clean data to evaluate the final 3D reconstruction quality, but they do not provide data necessary to evaluate the generative restoration model in isolation. Without this, it is difficult to determine if a method's improvement gain stems form better generative priors or others (such as better 3D initialization or other system-level factors).
> - **Standardized "Artifact Removal" Evaluation and Facilitate Generative 3D Reconstruction Research:** DL3DV-Res introduces the first standardized set of paired artifact frames and clean ground truth frames. This enables researchers to objectively measure how well a generative model can remove 3DGS artifacts while maintaining fidelity to the ground truth. High performance on this benchmark serves as a reliable indicator that the restoration model will lead to superior generative reconstruction results.
> - **Facilitate Generative NVS Research:** Moreover, for applications that require only artifact-free novel views without the need for a final 3D representation, DL3DV-Res can be a practical evaluation setting.
>
> ---
>
> **Q6: What are the training settings for your 3D Gaussian Splatting, including whether the points are initialized randomly or from colmap, and the learning rate for 3DGS, difix3D and your method?**
>
> All the methods initialize with the COLMAP point cloud in all the experiments and we filter and retain only the visible points from the input sparse training views (see line 808-809 of the original manuscript). The learning rate for 3DGS and our is the default setting such as position\_lr\_init=0.00016, position\_lr\_final=0.0000016, feature\_lr=0.0025, self.opacity\_lr=0.025, scaling\_lr=0.005 and rotation\_lr=0.001, and Difix3D we use their default setting that use learning rate=5e-6 and Adam optimizer.

---

> > ### Comment · Reviewer_YpNj · 2025-11-28
> >
> > Thanks for authors rebuttal, it addresses my most concerns, I decide to keep my initial score.

---

> > > ### Author Response · Authors · 2025-12-01
> > > **Thanks for Your Review**
> > >
> > > Thank you for your response! We are delighted to hear that our reply addressed your concerns and appreciate your recognition of our work.

---

### Official Review · Reviewer_C38R · 2025-10-27

**Soundness:** 3
**Presentation:** 2
**Contribution:** 3
**Rating:** 6
**Confidence:** 5

**Summary:**

This paper proposes GSFixer, a novel framework for improving 3DGS reconstruction quality in sparse-view scenes. Due to the underconstraint problem caused by sparse view input, traditional 3DGS is prone to geometric distortion and new-view artifacts. GSFixer inpaints these artifacts by introducing a reference-guided video diffusion model. During training, the model utilizes pairs of artifact-laden 3DGS rendered frames and high-quality GT frames. It also combines  DINOv2 feature extractor and 3D geometric features eature extractor from the sparse-view input to provide a hard constraint for the video diffusion model, thereby achieving semantically and geometrically consistent inpainting from new views.

**Strengths:**

1. **Originality-wise**: the paper proposes a novel framework for robust 3D reconstruction.
2. **Quality-wise**: the proposed method has combine Dinov2 and VGGT head to extract the 2D/3D features to fix the 3D scenes with gaussian splatting.
3. **Clarity-wise**: the manuscript is clearly written, with well-structured methodology, detailed explanations, and intuitive visualizations that enhance understanding.

**Weaknesses:**

1. **Limited generalization of camera trajectories.** The reference-guided trajectory strategy performs well on circumferential trajectories, but its adaptability and effectiveness on non-circumferential or sparse, discontinuous trajectories (such as extreme angles and non-closed-loop aerial photography) have not been fully verified. The model may be sensitive to the distribution of reference viewpoints and lack the ability to generalize to diverse trajectories. More exps and analysis in complex environments need to be conducted.

2. Concern about the dependency of poses and SfM initialization. The COLMAP results from all viewpoints were used as the initial point cloud, from which the sparse viewpoints required for training were selected. While this setup ensures high SfM initialization accuracy, it is inconsistent with real-world sparse scene applications. In real-world scenarios, input images are often few and viewpoints overlap less, directly using a sparse image set for COLMAP reconstruction may fail or significantly reduce accuracy and stability. I'm curious about the performance only with few input images without any eval images.

---
I have listed my concerns, and the score will be adjusted based on the author's response.

**Questions:**

Please refer to Weaknesses part.

---

> ### Author Response · Authors · 2025-11-25
> **Official Response by Authors**
>
> We would like to sincerely thank the reviewer for the valuable and constructive comments on our work. We take every comment seriously and hope our response can address the reviewer’s concerns.
>
> **Q1: Limited generalization of camera trajectories. The reference-guided trajectory strategy performs well on circumferential trajectories, but its adaptability and effectiveness on non-circumferential or sparse, discontinuous trajectories (such as extreme angles and non-closed-loop aerial photography) have not been fully verified. The model may be sensitive to the distribution of reference viewpoints and lack the ability to generalize to diverse trajectories. More exps and analysis in complex environments need to be conducted.**
>
> Thank you for the insightful suggestion. Following your recommendation, we conduct additional experiments on six scenes of AerialMegaDepth dataset [1], which features aerial photography scenes captured with discontinuous trajectories and extreme viewing angles. These aerial scenes are unseen in the training data with unseen  extreme viewpoint, and irregular captured trajectories, which are significantly more challenging than standard object-centric scenes of Mip-NeRF 360 dataset. Our results below demonstrate that GSFixer with the proposed reference-guided trajectory remains robust under diverse distribution of reference viewpoints. GSFixer consistently outperforms GenFusion in these complex environments further verify the effectivenesss of our method. The qualitative and quantitative comparison is presented in **Figure 11** and **Table 6** of our revised manuscript.
>
> [1] Aerialmegadepth: Learning aerial-ground reconstruction and view synthesis. CVPR 2025.
>
> | Method                                 | PSNR (9-view) | PSNR (12-view) | PSNR (24-view) | SSIM (9-view) | SSIM (12-view) | SSIM (24-view) | LPIPS (9-view) | LPIPS (12-view) | LPIPS (24-view) |
> |----------------------------------------|------------------|-------------------|-------------------|------------------|-------------------|-------------------|-------------------|--------------------|--------------------|
> | GenFusion                              | 11.23            | 11.43             | 13.07             | 0.399            | 0.411             | 0.477             | 0.578             | 0.562              | 0.520              |
> | GSFixer (ellipse trajectory)           | 11.99            | 12.57             | 13.56             | 0.406            | 0.433             | 0.495             | 0.570             | 0.553              | 0.499              |
> | GSFixer (reference-guided trajectory)  | 12.34            | 12.96             | 14.01             | 0.424            | 0.459             | 0.515             | 0.561             | 0.533              | 0.489              |
>
> ---
>
> **Q2: Concern about the dependency of poses and SfM initialization. The COLMAP results from all viewpoints were used as the initial point cloud, from which the sparse viewpoints required for training were selected. While this setup ensures high SfM initialization accuracy, it is inconsistent with real-world sparse scene applications. In real-world scenarios, input images are often few and viewpoints overlap less, directly using a sparse image set for COLMAP reconstruction may fail or significantly reduce accuracy and stability. I'm curious about the performance only with few input images without any eval images.**
>
> Thank you for the insightful suggestion. We agree with the reviewer that initializing with dense COLMAP points and filtering for visibility was to ensure high SfM initialization accuracy, and was designed for fair comparison with the previous works such as GenFusion (which also use this initialization). Follow your suggestion, we conduct additional experiments on Mip-NeRF 360 dataset where the initial point cloud are generated using only few images without any evaluation images. For extremely few images such as 3 views, COLMAP fails to generate points, so we conduct this evaluation using the 9-view setting. The results are presented below. While the use of sparse-only initialization results in a performance drop across all methods due to the lower quality of the initial geometry, GSFixer significantly outperforms the baselines. This demonstrates that our method has stronger generative capability and robustness than the baselines. We also provide some initial point cloud visualization and qualitative comparison in **Figure 12**, and quantitative comparison in **Table 7** of our revised manuscript.
>
> | Method     | PSNR | SSIM | LPIPS |
> |-----------|--------|--------|---------|
> | 3DGS      | 13.82  | 0.287  | 0.559   |
> | Difix3D+  | 14.60  | 0.315  | 0.500   |
> | GenFusion | 14.71  | 0.362  | 0.643   |
> | GSFixer   | 16.57  | 0.383  | 0.503   |

---

> > ### Comment · Reviewer_C38R · 2025-11-25
> >
> > Thank you for the author's reply. The author's reply has addressed my concerns, and I tend to accept this work.
> > I maintain my current rate.

---

> > > ### Author Response · Authors · 2025-11-27
> > > **Official Comment by Authors**
> > >
> > > Thank you for your response! We are delighted to hear that our reply addressed your concerns and appreciate your recognition of our work.

---

### Official Review · Reviewer_g5Ny · 2025-10-31

**Soundness:** 3
**Presentation:** 3
**Contribution:** 2
**Rating:** 4
**Confidence:** 3

**Summary:**

This work proposes GSFixer designed to improve the quality of 3D Gaussian Splatting (3DGS) reconstructions from sparse input views.

GSFixer: A generative reconstruction pipeline integrating a reference-guided video restoration model and trajectory sampling strategy.

Dual Conditioning: Uses both 2D semantic and 3D geometric features from reference views to guide artifact correction.

DL3DV-Res Benchmark: A new dataset for evaluating 3DGS artifact restoration.

Performance: Good results in artifact restoration and sparse-view reconstruction across multiple benchmarks.

**Strengths:**

I think the most original contribution is the reference-guided video restoration model, which conditions a video diffusion model on both 2D semantic features (via DINOv2) and 3D geometric features (via VGGT) extracted from the input sparse views. Different from previous works, the condition is multimodal.

The introduction of the RGT strategy is a clever way to refine the 3DGS systematically.

**Weaknesses:**

The scales, dimensions, and information densities of the geometric features (VGGT) and semantic features (DINOv2) may differ, and improper fusion may result in the weakening of one of the features. How to confirm that the fusion is correct for both features? Did not see the ablation studies of this part.

**Questions:**

Could you have some attention visualizations to confirm that the fusion works?

---

> ### Author Response · Authors · 2025-11-25
> **Official Response by Authors**
>
> We would like to sincerely thank the reviewer for the valuable and constructive comments on our work. We take every comment seriously and hope our response can address the reviewer’s concerns.
>
> **Q1 & Q2: The scales, dimensions, and information densities of the geometric features (VGGT) and semantic features (DINOv2) may differ, and improper fusion may result in the weakening of one of the features. How to confirm that the fusion is correct for both features? Did not see the ablation studies of this part. Could you have some attention visualizations to confirm that the fusion works?**
>
>
> We would like to sincerely thank the reviewer for the valuable and constructive comments on our work. We provide our original manuscript, theoretical analysis, combined with the new attention heatmap and PCA visualizations to confirm that the fusion is correct.
>
> ### 1. Experimental Confirmation (Original Ablation Studies)
>
> We would like to clarify that we provided the necessary ablation studies of the feature conditions in **Table 4 and Figure 6** (Section 4.2, line 428-463 of our original manuscript), which **confirm that the fusion is useful from experiments**.
>
> - **Quantitative Ablation (Table 4)**: Our full model, utilizing the fusion of 3D geometric (VGGT) and 2D semantic (DINOv2) features, significantly outperforms ablation variants that use only 3D tokens or only 2D tokens across all metrics.
> - **Qualitative Ablation (Figure 6)** show that removing the 3D condition leads to poor 3D consistency with the reference views, resulting in unstable geometry. Removing the 2D condition causes the model to fail in restoring semantically plausible details.
>
> ### 2. Theoretical Confirmation
>
> The potential issue of scales, dimensions, and information densities of the geometric features (VGGT) and semantic features (DINOv2) can be handled through our fusion implementation:
>
> - **Normalization:** the geometric and semantic features are processed separately through MLPs projectors (linear and normalization layers). The normalization layers ensure that both geometric and semantic features are brought to the same magnitude scale.
> - **Dimension Mapping:** the linear layers map the dimensions of both features to match the dimension of the diffusion model's intermediate layer.
> - **Adaptive Weighting:** after passing through the projectors, the information density of the geometric and semantic features, will be adaptively weighted after our fine-tuning process.
>
> ### 3. More Visual Confirmation (Attention Heatmap and PCA Visualizations)
>
> Following your suggestion, we **provide some attention visualization in Figure 9 and Figure 10** of our revised manuscript to confirm that the fusion works.
>
> - **Attention Heatmap: Figure 9** visualizations demonstrate how different features prioritize information in the reference images, leading to distinct patterns for DINOv2, VGGT, and the Fused features. From the visualization, we can observe that the VGGT attention heatmap shows it activation primarily concentrated along sharp geometric details, such as edges, corners, boundaries, while the DINOv2 attention heatmap exhibits high activation is broad and continuous across the entire area of recognized objects. The Fused feature map combines the strengths of both VGGT and DINOv2: high activation over the entire semantic area (like DINOv2), but with the activation map being enhanced and refined by geometric boundaries (like VGGT). This indicates the fusion successfully captures a holistic representation that is robust to global context while retaining local geometry awareness.
>
> - **PCA Visualization:** In addition, we further provide Principal Component Analysis (PCA) visualization in **Figure 10** of our revised manuscript to confirm that the fusion works. We reduce the high-dimensional feature vectors of each feature to a three-dimensional space, which is then directly mapped to the RGB color space to visualize semantic clustering. This is widely used in the analysis of DINOv2 feature upsampler work such as LoftUp [1]. From the visualization, we can observe that PCA visualization of DINOv2 features shows large, contiguous blocks of color that align with foreground objects or background regions, demonstrating its ability to capture semantic clusters. The fused features exhibits the desired combination: stable semantic clusters (inherited from DINOv2) whose boundaries are now refined with the detail geometry (inherited from VGGT).
>
> [1] LoftUp: Learning a coordinate-based feature upsampler for vision foundation models. ICCV 2025.

---

### Official Review · Reviewer_4bNP · 2025-11-01

**Soundness:** 2
**Presentation:** 2
**Contribution:** 1
**Rating:** 4
**Confidence:** 5

**Summary:**

The paper proposes GSFixer, which uses a DiT-based video diffusion model (CogVideoX) to “repair” artifact-ridden 3DGS novel views, conditioning the diffusion on 2D semantic tokens (DINOv2) and 3D geometric tokens (VGGT) extracted from sparse reference images. The restored frames are then distilled back into 3DGS in an iterative loop, and a reference-guided trajectory is introduced to balance view quality and angular coverage. The authors also release DL3DV-Res, a benchmark of artifact frames rendered from low-quality 3DGS, and report improvements over GenFusion and Difix3D+ on restoration and sparse-view NVS. Results on DL3DV and Mip-NeRF-360 show mostly modest gains in PSNR/SSIM/LPIPS gains, with ablations indicating that both 2D and 3D tokens are beneficial.

**Strengths:**

**Task framing + benchmark.** Explicitly casting “artifact restoration” for sparse-view 3DGS and introducing **DL3DV-Res** gives a concrete testbed; the task is well motivated, and the dataset construction is described. Reported scores show clear, if not dramatic, gains over prior generative baselines on this benchmark.

- **Dual conditioning (2D + 3D tokens).** Conditioning the video diffusion on DINOv2 (semantics) and VGGT (geometry) is a coherent way to push consistency to the fixed frames, and the injection via cross-attention is straightforward. The ablation results show that the full model outperforms the “w/o 2D” and “w/o 3D” variants across all metrics.

- **Reference-guided trajectory.** The paper identifies a hybrid camera path that balances interpolation quality with spherical coverage. Although the gains are small, the trajectory is at least validated with both qualitative and quantitative comparisons.

**Weaknesses:**

**Incremental relative to recent generative NVS systems** - The method largely leverages known techniques, including video diffusion restoration, iterative distillation back to the 3D representation, and simple camera path sampling, and is very close in spirit to GenFusion/Difix3D+. ** The paper’s own related-work section positions it as a variation rather than a conceptual step change. The only new contribution is the type of conditioning signal used in the fix step, making this an incremental improvement to an existing framework rather than a new one. The novelty claim boils down to adding dual token conditioning and a minor trajectory tweak.
- **Trajectory ablation has small effect sizes** (e.g., ≤0.2 dB PSNR) over usual elliptical trajectories, making it unclear whether the complexity is warranted.
- **Under-analyzed compute footprint** - The core model is CogVideoX-5B with ~50 diffusion steps, plus extra encoders (BLIP, DINOv2, VGGT) at inference/conditioning; there’s no runtime/memory breakdown or wall-clock profiling, and the method’s practicality is relegated to a brief limitations section. For a system paper, this lacks the expected engineering transparency.

**Questions:**

N/A

---

> ### Author Response · Authors · 2025-11-25
> **Official Response by Authors -- Part 1**
>
> We would like to sincerely thank the reviewer for the valuable and constructive comments on our work. We take every comment seriously and hope our response can address the reviewer’s concerns.
>
> **Q1: Incremental relative to recent generative NVS systems - The method largely leverages known techniques, including video diffusion restoration, iterative distillation back to the 3D representation, and simple camera path sampling, and is very close in spirit to GenFusion/Difix3D+. The paper’s own related-work section positions it as a variation rather than a conceptual step change. The only new contribution is the type of conditioning signal used in the fix step, making this an incremental improvement to an existing framework rather than a new one. The novelty claim boils down to adding dual token conditioning and a minor trajectory tweak.**
>
> Thank you for highlighting these relevant works. We believe that our method has non-trivial and meaningful contributions to the 3D reconstruction field including:
>
> - **Addressing the "Consistency Gap" via Reference-Guided Video Priors:** Existing works lack designs for enforcing the consistency between the generated views and the given sparse views. For instance, Difix3D+ concatenates a noise frame with its nearest clean view and applies an *image* diffusion model for per-frame correction, which inherently lacks the cross-frame coherence. GenFusion adopts video diffusion model but relies on weak guidance (CLIP embedding), which often leads to noticeable misalignment with the input views. In contrast, we are *the first* to explicitly formulate artifact view restoration as a reference-guided video generation problem. Our GSFixer takes two clean views along with noisy renders in between, ensuring that the generated sequence remains tightly anchored to the appearance of the input views. Beyond simply conditioning on clean images, we further explicitly inject rich information from clean reference views into the video diffusion model through cross-attention. This forces the model to ground its generation in reliable input signals rather than hallucinating plausible yet inconsistent results, effectively resolving the inconsistency gap that remains unaddressed in existing literature.
>
> - **Novel Triple-Conditioning:** Beyond simple input concatenation, considering the artifacts occurs in the 2D image space and originates from suboptimal 3D representations, we propose a triple-conditioning mechanism that incorporates both 2D  and 3D features extracted from clean reference views, thereby enhancing the model’s understanding of both domains. This triple-conditioning effectively guides the video diffusion process, achieving both semantically coherent and 3D consistent restoration results. To our knowledge, we are the *first* to leverage geometry (VGGT) and semantic features (DINOv2) of the input views for artifact restoration, which remains underexplored in existing literature.
>
> - **DiT Architecture and Benchmarking:** We are the *first* to tailor advanced Diffusion Transformer (DiT) video diffusion architecture for this task, moving beyond the U-Net paradigms used in previous work.  The design is empirically grounded: as detailed in manuscript (refer to Sec. 3, Fig. 3, Tab. 1 & 4), we extensively explore the architectural design space to effectively adapt the image-to-video model to video-to-video restoration setting and augment each DiT block by adding cross-attention layer after the 3D attention layer,  which significantly outperforms the baselines (+2.16 in PSNR). In addition, we introduce the DL3DV-Res benchmark, providing the community the *first* standardized evaluation protocol for generative 3DGS restoration. This benchmark facilitate more comparable future research in generative NVS and reconstruction.
>
> ---
>
> **Q2: Trajectory ablation has small effect sizes (e.g., ≤0.2 dB PSNR) over usual elliptical trajectories, making it unclear whether the complexity is warranted.**
>
> Thank you for your thoughtful comment. The improvements brought by our reference-guided trajectory are comparable to those reported in the existing works. For instance, the core ablation study of Difix3D+ (Table 4 in their paper) reports performance gains for their main components (“Difix” and “Difix3D+”) each contribute only about 0.1 dB and 0.2 dB PSNR, respectively. Moreover, we clarify that the complexity of the reference-guided trajectory refers to the design of the path sampling strategy. Computing these sampling paths takes less than one second, which is identical to the standard elliptical trajectory. Since both strategies sample the same number of novel views, there is no extra rendering or diffusion inference cost.

---

> ### Author Response · Authors · 2025-11-25
> **Official Response by Authors -- Part 2**
>
> **Q3: Under-analyzed compute footprint - The core model is CogVideoX-5B with ~50 diffusion steps, plus extra encoders (BLIP, DINOv2, VGGT) at inference/conditioning; there’s no runtime/memory breakdown or wall-clock profiling, and the method’s practicality is relegated to a brief limitations section. For a system paper, this lacks the expected engineering transparency.**
>
> Thank you for your valuable suggestion. We agree that engineering transparency is crucial. In response, we have added a detailed breakdown of runtime and memory usage for GSFixer and key baselines (Difix3D+ and GenFusion), including both the restoration stage and the full reconstruction pipeline. All results in the below table are averaged over the DL3DV-Res benchmark (restoration) and Mip-NeRF 360 (3, 6, and 9-view reconstruction), evaluated on a single NVIDIA H100 GPU.
>
> - **Runtime:** while our per-frame restoration time (4.32s) is higher due to the 50-step CogVideoX-I2V-5B based model, our total scene reconstruction time  remains comparable to Difix3D+. This is because Difix3D+ utilizes a fast one-step image diffusion model (0.75s/frame) but requires 29 fix-and-distill iteration to achieve the final results, while our GSFixer use two cycles in all experiments. Our work focuses on solving the inconsistency issues to improve the performance, and future work (e.g., caching [1], few-step distillation [2]) can accelerate GSFixer without modifying the pipeline.
>
> - **Memory Cost:** we acknowledge the higher memory footprint necessitated by the advanced video diffusion model (CogVideoX-I2V-5B) and extra encoders (VGGT, DINOv2, BLIP). However, this cost is justified by the significant improvements in fidelity and consistency (e.g., +2.16 dB PSNR over GenFusion and Difix3D+ on DL3DV-Res). We also note that the recent video diffusion model (Wan-1.3B) achieves comparable generation quality with much lower memory requirements and can be adopted in future extensions of GSFixer without modifying the pipeline.
>
> In addition, our work, including code, model checkpoints, and benchmarks will be released to enable full engineering transparency.
>
> [1] Adaptive caching for faster video generation with diffusion transformers. ICCV 2025.
>
> [2] Self Forcing: Bridging the Train-Test Gap in Autoregressive Video Diffusion. ArXiv 2025.
>
> | Method       | **Restoration**            |                | **Reconstruction**               |                |             |
> |--------------|----------------------------|----------------|----------------------------------|----------------|-------------|
> |              | Time (s / frame)           | Memory (GB)    | Time (min / scene)               | Memory (GB)    | Iteration   |
> | DifFix3D+    | 0.75                       | 21.9           | 68                               | 25.3           | 29          |
> | GenFusion    | 1.09                       | 19.3           | 39                               | 28.2           | 5           |
> | GSFixer      | 4.32                       | 25.6           | 72                               | 49.8           | 2           |

---

### Author Response · Authors · 2025-11-25
**General Response**

We thank all reviewers for their detailed reviews and suggestions!

We have updated the manuscript with the following revisions based on the reviewers' suggestions. All revisions in the updated manuscript are highlighted in red:

**Add more experiments**
- Add qualitative attention heatmap and PCA visualiztion about VGGT, DINOv2 and Fused features (reviewer g5Ny, Appendix E line 959-1015, Figure 9 and Figure 10).
- Add experimental results on complex aerial photography scenes (reviewer C38R-Q1, Appendix F, line 1020-1065, Figure 11 and Table 7).
- Add experimental results about SfM initialization with few images (reviewer C38R-Q2, Appendix G, line 1072-1079, Figure 12 and Table 8)
- Add experimental results about conditioning on depthmap output of VGGT (reviewer YpNj-Q1, Appendix H, line 1115-1022, Figure 13 and Table 9).
- Add experimental results about Difix3D+ with different number of reference views (reviewer YpNj-Q2, Appendix I, line 1127-1060, Figure 14 and Table 10).
- Add experimental results about Difix3D+ with and without post-rendering refinement step (reviewer YpNj-Q4, Appendix I, line 1162-1067, Figure 15 and Table 11).

**Add more discussion and detials**
- Add detail results about runtime and memory cost (reviewer 4bNP-Q3, Appendix D, line 915-954, Table 6).
- Expand related works in Sec. 2. Discuss with ReconX and FlowR. (reviewer YpNj-Q2-Q3, line 136-139).

Thanks again for all the effort and time, and we look forward to further discussions if there are any more questions.

---

> ### Author Response · Authors · 2025-12-01
> **Further Summary**
>
> Dear reviewers and AC,
>
> We sincerely thank all reviewers for their valuable feedback and encouraging comments regarding our **novel and effective reference-guided video restoration model**, which introduces features from **3D (VGGT) and 2D (DINOv2) as additional conditioning signals** for video diffusion (Reviewer 4bNP, g5Ny, C38R, YpNj), **novel reference-guided trajectory** (Reviewer 4bNP, g5Ny, YpNj), **task-specific benchmarks** (Reviewer 4bNP), **comprehensive qualitative and quantitative evaluation** (Reviewer 4bNP, C38R, YpNj) and **clarity and quality of writing** (Reviewer C38R, YpNj).
>
> Before the discussion period was interrupted, two reviewers provided updated feedback:
>
> Reviewer C38R maintained the initial score at 6, noting that our rebuttal addressed the concerns and tend to accept this work.
> Reviewer YpNj maintained the initial score at 6, confirming that our rebuttal addressed their most concerns.
>
> For the two reviewers who did not provide later comments, we have already provided detailed rebuttals addressing their concerns:
>
> Reviewer 4bNP's primary concerns regarding system novelty (Response 4bNP-Q1) and compute footprint (Response 4bNP-Q3 and Appendix D, Table 6).
> Reviewer g5Ny's concern regarding the confirmation of feature fusion (Response g5Ny-Q1, Table 4 and Figure 6, Appendix E, Figures 9 and 10).
>
> We sincerely hope that our analysis and clarifications address all concerns.
>
> Thank you for your time and consideration.
>
> Best regards,
>
> Authors

---

### Author Response · Authors · 2025-11-27
**Thanks for Your Review**

Dear Reviewers,

Thanks again for the constructive comments and the time you dedicate to the paper! Many improvements have been made according to your suggestions and comments. We also updated the manuscript and summarized the changes below. We hope that our responses and the revision will address your concerns.

Since the discussion is about to close, we would be grateful if you could take some time to review our responses and the revised manuscript. We are glad to follow up with your further comments.

Thanks a lot again, and with sincerest best wishes

Submission 4833 Authors

---

### Meta-Review · Area_Chair_eLAP · 2026-01-07

**Summary:**

Reviewer 4bNP and Reviewer YpNj viewed the contribution as somewhat incremental relative to recent generative NVS systems, with only modest gains and small effect sizes, particularly for the proposed trajectory design. Reviewer g5Ny was concerned that the paper does not sufficiently analyze how the 2D (DINOv2) and 3D (VGGT) conditioning features interact or whether their fusion is balanced. Reviewer 4bNP also noted that the method’s computational footprint and practicality are under-analyzed. Finally, Reviewer C38R questioned the generalization of the method to more realistic sparse-view settings and highlighted the dependence on strong SfM initialization.

**Reviewer Concerns:**

It appears that the reviewers’ concerns have been addressed. However, based on my own reading of the paper, the contributions seem incremental and do not clearly meet the acceptance threshold (which is consistent with the reviewers’ scores). I would encourage the authors to further strengthen the work and consider submitting it to a future conference.

**Reviewer Scores:**

Reviewer 4bNP: 4
Reviewer g5Ny: 4
Reviewer C38R: 6
Reviewer YpNj: 6

Reviewers C38R and YpNj are satisfied with the rebuttal and have chosen to maintain their original scores. Based on the authors’ responses, it appears that the concerns raised by Reviewers 4bNP and g5Ny have also been addressed to some extent. It is likely that they will either keep their current scores or raise them slightly.

---

### Decision · Program_Chairs · 2026-01-26

Reject